# Correspondence between cerebral glucose metabolism and BOLD reveals relative power and cost in human brain

Ehsan Shokri-Kojori [1], Dardo Tomasi [1], Babak Alipanahi [2], Corinde E. Wiers [1], Gene-Jack Wang [1] & Nora D. Volkow [1,3]

The correspondence between cerebral glucose metabolism (indexing energy utilization) and synchronous fluctuations in blood oxygenation (indexing neuronal activity) is relevant for neuronal specialization and is affected by brain disorders. Here, we define novel measures of relative power (rPWR, extent of concurrent energy utilization and activity) and relative cost (rCST, extent that energy utilization exceeds activity), derived from FDG-PET and fMRI. We show that resting-state networks have distinct energetic signatures and that brain could be classified into major bilateral segments based on rPWR and rCST. While medial-visual and default-mode networks have the highest rPWR, frontoparietal networks have the highest rCST. rPWR and rCST estimates are generalizable to other indexes of energy supply and neuronal activity, and are sensitive to neurocognitive effects of acute and chronic alcohol exposure. rPWR and rCST are informative metrics for characterizing brain pathology and alternative energy use, and may provide new multimodal biomarkers of neuropsychiatric disorders.

[1] National Institute on Alcohol Abuse and Alcoholism, National Institutes of Health, Bethesda 20892 MD, USA. [2] Mountain View, California, USA. [3] National Institute on Drug Abuse, National Institutes of Health, Rockville 20892 MD, United States. Correspondence and requests for materials should be addressed to E.S-K. (email: ehsan.shokrikojori@nih.gov)

The human brain has markedly complex and diverse structural characteristics that have evolved in accordance with the functional specialization of brain regions[1]. Notably, larger brain mass, indicating higher cognitive capacity, has been associated with higher energetic cost across species[2]. In this respect, differences in regional morphometry, axonal and dendritic density, glia-to-neuron ratio, neurotransmitter distribution, and active metabolic pathways, have led to different functional outcomes for different brain regions, including variations in baseline glucose metabolism[3,4] and differences in neuroglial activity levels[5,6]. It has been shown that regional glucose metabolism is coupled with activity levels at rest[7,8] and task conditions[9–11]. In this relation, indirect measures of neuronal activity, such as functional magnetic resonance imaging (fMRI) measures of functional connectivity[12,13] or magnetic resonance spectroscopy measures of glutamatergic function[14], have been associated with regional brain glucose metabolism, wherein high and low neuronal activity demand were associated with high and low metabolic supply, respectively. Compelling evidence also indicates that brain regions may differ in their activity demand and metabolic supply associations[15]. For example, glucose metabolism (brain's main energy supply) may exceed neuronal activity demand when less efficient glucose metabolic pathways, such as aerobic glycolysis, are favored in a given region compared to the rest of the brain[16]. Contrastingly, glucose metabolism may fall behind neuronal demand when relative to other regions, there is more reliance on the Krebs cycle, or when alternatives to glucose such as ketone bodies are metabolized as substrates for energy generation[17,18].

Emerging evidence suggests that the coupling between neuroglial demand and energy supply entails a bidirectional association[19]; however, the spatiotemporal dynamics of these associations remain to be further explored. Variations in how energy is supplied (and metabolized) in different brain regions (spatially)[20] and under different stimulation and physiological conditions (temporally)[21] are of high relevance in our understanding of brain physiology[22], development[20], cognitive abilities[23], and neuropsychiatric disorders[24]. There are marked regional differences in glucose metabolism[3,25] and in fMRI measures of brain activity[6,26,27] during resting state that are positively associated across regions[12,13]. However, without accounting for underlying brain activity, regional differences in glucose metabolism are hard to interpret. Interestingly, the level of correspondence between glucose metabolism and neuroglial activity has been considered as a marker of functional specialization[16], and could be helpful for inferring alternative energy use vs. activation of different metabolic pathways. Here we propose an approach to quantify match and mismatch between measured metabolic supply and the observed level of activity across the brain and assessed whether this quantification is relevant for studying distinct energetic characteristics of brain regions and networks. For this purpose, we measured cerebral metabolic rate of glucose (CMRglc, indexed by[18]F-flurodeoxyglucose; fluorodeoxyglucose-positron emission tomography (FDG-PET), see Methods) and synchronous fluctuations in the blood oxygenation level dependent (BOLD; measured by fMRI and indexed by local functional connectivity density: lFCD, see Methods) during resting state. We studied two main (unit-free and generalizable) dimensions of associations. The first dimension captured the positive association between glucose utilization and neuroglial activity and was labeled relative power (rPWR), which represented the level of concurrent metabolic need and observed activity, relative to the rest of the brain. The second dimension captured the deviation between glucose utilization and neuroglial activity and was labeled relative cost (rCST), which represented the extent to which glucose metabolic needs exceed (or fall behind) the observed activity, relative to the rest of the brain. As

in principal component analysis, when there is complete correspondence between measured neuroglial activity and glucose utilization across regions, all the common variance will be accounted for by the rPWR dimension. But, more deviation between observed neuroglial activity and glucose utilization[16,20,21,28] (i.e., disproportional neuroglial activity and glucose utilization) across regions would result in higher variance accounted for by the rCST dimension (Fig. 1, see Methods).

Here we perform a series of experiments and analyses in two independent cohorts. In cohort-1 ($n = 28$ healthy participants) with high-resolution FDG-PET and fMRI, we compute voxelwise measures of rPWR and rCST. We test the hypothesis that different brain networks have distinct rPWR and rCST signatures. We use this characterization of the brain's lFCD-CMRglc dynamics (indexing important components of neuronal activity demand and metabolic supply) and classify the brain into major segments based on rPWR and rCST. We show the generalizability of rPWR and rCST to alternative measures of metabolic supply (i.e., cerebral blood flow: CBF) and measures of neuronal activity (i.e., fractional amplitude of low-frequency fluctuations: fALFF). We also assess effects of temporal signal-to-noise ratio (tSNR) and brain morphometry on rPWR and rCST. In cohort-2 ($n = 40$) with FDG-PET and fMRI, we test the sensitivity of rPWR and rCST to acute and chronic alcohol exposure, which affect brain glucose metabolism[9,29,30] and neuronal activity[31,32], in light drinkers (LDs, $n = 24$) and heavy drinkers (HDs, $n = 16$). We propose multimodal measures of rPWR and rCST to study regional variations in the correspondence between glucose metabolism and measures of functional activity, with potential implications for characterizing neuropsychiatric diseases. Please refer to Table 1 for the list of acronyms.

## Results

**Voxelwise rPWR and rCST.** Spatial distributions of lFCD and CMRglc are highlighted in Fig. 1a–c and Supplementary Figs. 1a and 2, delineating regional variability in these measures while showing an overall positive association between lFCD and CMRglc across the brain regions (cohort-1, $n = 28$). To quantify the regional differences in the coupling between lFCD (indexing synchronous BOLD fluctuations and related to activity demand) and CMRglc (indexing glucose metabolic supply), we defined measures of rPWR and rCST. While rPWR captured the level of concurrent lFCD and CMRglc, rCST captured the mismatch between lFCD and CMRglc, relative to the rest of the brain. In a two-dimensional map of (mean-variance-normalized) lFCD-CMRglc (Fig. 1e), we performed an orthogonal transformation with a counterclockwise $\pi/4$ (45°) rotation of axes (Fig. 1e; see Methods) and defined an rPWR axis along which the positive end indicated high CMRglc associated with high lFCD and the negative end indicated low CMRglc associated with low lFCD. Perpendicular to the rPWR axis, we define an rCST axis (Fig. 1e) along which the positive end indicated high CMRglc associated with low lFCD and the negative end indicated low CMRglc associated with high lFCD. Fig. 1e shows a hypothetical model with relatively equal distribution of voxels along the four identified quadrants and with no apparent association between the measures. A positive correlation between lFCD (measure of activity) and CMRglc (measure of metabolic supply) indicates that more voxels are associated with high- and low-rPWR quadrants than high- and low-rCST quadrants. Voxels contributing the most to rPWR and rCST variability (in the hypothetical model) are highlighted in Fig. 1d, f and Supplementary Fig. 3, respectively. We found marked regional differences in rPWR and in rCST (Fig. 1g, h and Supplementary Fig. 3) that were driven by match and mismatch between lFCD and CMRglc

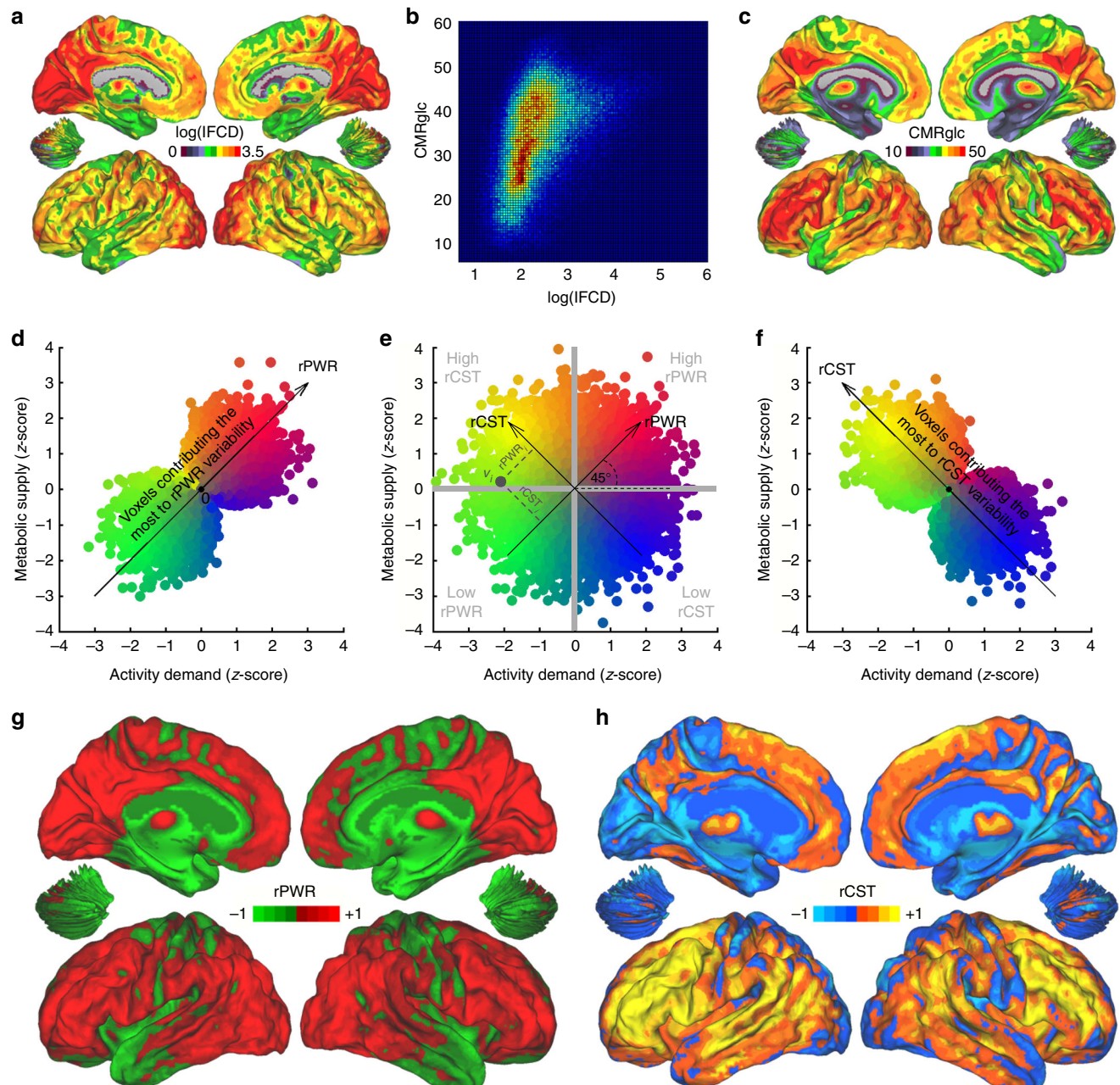

**Fig. 1** Voxelwise relative power (rPWR) and relative cost (rCST) (cohort-1, $n = 28$). **a** Average logarithm of local functional connectivity density (log(lFCD)) map for cohort-1 ($n = 28$) subjects highlighting precuneus and visual cortex as the most locally connected regions. **b** COMET (connectivity-metabolism) map. A two-dimensional histogram of log(lFCD) versus cerebral metabolic rate of glucose (CMRglc) highlighting the frequency of log(lFCD) and CMRglc association pairs ($r = 0.44$, $p < 0.00001$, 139,269 voxels). Red to blue indicate high to low-frequency counts, respectively. **c** Average CMRglc map highlighting precuneus and lateral frontal lobe as the most metabolically demanding regions. **d**–**f** rPWR and rCST were calculated by a $\pi/4$ (45°) rotation along (mean-variance normalized) log(lFCD) and CMRglc axes. **e** A hypothetical presentation of (mean-variance normalized) activity demand versus metabolic supply (not to be confused with part **b** shown without mean-variance normalization) with each circle representing one brain voxel. Yellow-colored voxels correspond to higher rCST, blue to lower rCST, red to higher rPWR, and green to lower rPWR. For a representative voxel $v_i$ (dark gray circle), $rPWR_i$ and $rCST_i$ are shown on the plot. For $v_i$, rPWR is negative and rCST is positive. For visual demonstration purposes, voxels contributing the most to rPWR variability (**d**) and rCST variability (**f**) are highlighted. For this purpose, highlighting was performed by multiplying the radius of each voxel (in polar coordinate system) by its corresponding absolute rPWR (**d**) or absolute rCST (**f**). Group-average rPWR (**g**) and rCST (**h**) maps. Note the color scale in **g** resembles that used for the hypothetical voxels along the rPWR axis in **d** and the color scale in **h** resembles that used for the hypothetical voxels along the rCST axis in **f**. Also see Fig. 2 (and Supplementary Table 8) for regional differences

(Fig. 1a, c). Regions with higher rPWR included major sections of visual, parietal, and frontal cortices, putamen, caudate, and medial–dorsal nucleus of the thalamus (Fig. 1g, Supplementary Table 1). Conversely, parts of the anterior and posterior cerebellar lobes, medial–frontal, and precentral gyri, hippocampus and

parahippocampal gyrus, and ventral anterior and lateral nuclei of the thalamus had lower rPWR (Fig. 1g, Supplementary Table 1). Many brain regions with higher rPWR had higher rCST (Fig. 1g, h). Higher rCST regions included inferior, middle, superior frontal, precentral, and postcentral gyri, as well as insula,

**Table 1 List of acronyms used in the text and supplementary material**

| Acronym | Description |
| --- | --- |
| ABND | average between network distance |
| AWND | average within network distance |
| ALC | alcohol (condition) |
| ALFF | amplitude of low-frequency fluctuations |
| ANOVA | analysis of variance |
| BOLD | blood oxygenation level dependent |
| CB | cerebellum |
| CBF | cerebral blood flow |
| CM | center of mass |
| CMRglc | cerebral metabolic rate of glucose |
| COMET | connectivity-metabolism |
| fALFF | fractional amplitude of low-frequency fluctuations |
| FDG | $^{18}$F-flurodeoxyglucose |
| fMRI | functional magnetic resonance imaging |
| FWE | family wise error |
| HD | heavy drinker |
| ICC | intraclass correlation |
| LD | light drinker |
| lFCD | local functional connectivity density |
| M | mean |
| MNI | Montreal neurological institute |
| MV | medial visual |
| NSI | network segregation index |
| PC | principal component |
| PET | positron emission tomography |
| PLC | placebo (condition) |
| PWI | perfusion-weighted imaging |
| rCST | relative cost |
| ROI | region of interest |
| rPWR | relative power |
| SD | standard deviation |
| TE | echo time |
| TR | repetition time |
| tSNR | temporal signal to noise ratio |

putamen, and middle, and superior temporal gyri. In contrast, caudate, cerebellum (CB), limbic lobe, midbrain, pons, and ventral anterior and lateral thalamic nuclei had lower rCST (Fig. 1h, Supplementary Table 2).

**Resting state network and individual differences**. Figure 2a, b shows individual differences in lFCD and CMRglc within 10 predefined resting-state networks[33]. Plotting these values against each other showed low segregation of these networks when considering individual differences (Fig. 2c). Similarly, we compared individual differences in rPWR and rCST of these 10 networks (Fig. 2d, e). There was a main effect of network for both rPWR ($F(9, 243) = 197.86$, $p < 0.0001$) and rCST ($F(9, 243) = 203.51$, $p < 0.0001$) measures. Specifically, all network pairs had either significantly different rCST or significantly different rPWR ($p_{FWE} < 0.05$, Sidak), except the left and right frontoparietal networks ($p_{FWE} > 0.7$, Sidak). In addition, differences in rPWR or in rCST in some networks were not significant (e.g., rPWR between medial–visual (MV) and default mode networks or rCST between the occipital pole (OP) and CB networks; $p_{FWE} > 0.7$, Sidak). Highest rPWR corresponded to the MV and default mode networks, whereas lowest rPWR corresponded to the CB network. The left and right frontoparietal networks had the highest rCST, whereas the MV and CB networks had the lowest rCST. Since rPWR and rCST were calculated from mean-variance normalized lFCD and CMRglc, both measures contributed to variance in rPWR and rCST (see Methods). In addition, all networks had

significantly different rPWR compared to rCST ($p < 0.02$, paired $t$-test).

**Network segregation analysis**. Figure 2f shows subject-level averages of rPWR and rCST when plotted against each other, which in contrast to Fig. 2c, highlighted consistency of rPWR and rCST values across subjects for each network relative to other networks. This resulted in better segregation of networks based on rPWR and rCST properties than lFCD and CMRglc measures (Fig. 2c, f). We quantified the level of segregation of brain networks in the two spaces (i.e., lFCD-CMRglc versus rPWR-rCST) using a network segregation index (NSI). Specifically, in a given two-dimensional space and for each brain network, NSI was defined as the ratio of the average of between network distances to the average of within network distances (see Methods). As expected, in the rPWR-rCST space we found significantly higher NSIs (NSI: $M = 4.34$, SD = 1.46, Fig. 2f) than in the lFCD-CMRglc space (NSI: $M = 1.07$, SD = 0.66, Fig. 2c) across the 10 networks ($p = 0.002$, Wilcoxon's signed rank test).

**$k$-means clustering**. There were notable differences in regional distributions of rPWR and rCST (Fig. 3a). The largest differences between rPWR and rCST maps were in the visual cortex, which despite having high rPWR had low rCST, while many temporal and limbic regions showed the opposite pattern (Supplementary Table 3). Variations in the regional rPWR and rCST measures motivated segmenting the brain into groups of voxels, each with most similar rPWR and rCST than the rest of the brain. A $k$-means clustering approach identified four reproducible clusters of voxels (see Methods) from the across-subject average of rPWR and rCST measures (Fig. 3b, c and Supplementary Fig. 4). The clusters corresponded to: higher rPWR, lower (to intermediate) rCST (red); intermediate rPWR, higher rCST (yellow); lower rPWR, intermediate rCST (blue); and lower rPWR, lower rCST (green) regions (Fig. 3b–d). Projection of these clusters into the across-subject average of lFCD-CMRglc map (Fig. 3c) showed that high lFCD and high CMRglc corresponded to the red cluster, whereas the other three clusters showed low (green), intermediate (blue), and high (yellow) CMRglc for relatively low lFCD levels, respectively. Voxels corresponding to each of the clusters primarily highlighted sensorimotor (blue), cerebellar-limbic (green), visual (red), and frontoparietal (yellow) regions (Fig. 3d), but these segments also included a range of other regions throughout the brain (Supplementary Tables 4–7).

**Generalizability and effects of tSNR and brain morphometry**. We further assessed whether rPWR and rCST are generalizable to alternative measures of neuronal activity (i.e., fALFF) and metabolic supply (i.e., CBF with perfusion-weighted imaging (PWI)) and studied the extent to which they are sensitive to measurement noise (tSNR) and brain morphometry. In sum, there were high correlations between regional differences in CMRglc and CBF (PWI) and between regional differences log (lFCD) and fALFF (Supplementary Fig. 5). There was excellent agreement between rPWR based on lFCD-CMRglc, and rPWR based on fALFF-CBF (PWI) (Supplementary Fig. 6). Same effect was observed for rCST (Supplementary Fig. 7). Across regions, rPWR and rCST did not amplify the effects of tSNR and brain morphometry on lFCD and on CMRglc (Supplementary Tables 8–12). See Supplementary Results for more details.

**Effects of alcohol on regional rPWR and rCST**. We assessed the sensitivity of rPWR and rCST metrics to acute and chronic alcohol exposure in a separate cohort ($n = 40$; see Methods). Figure 4a–d shows the connectivity metabolism (COMET) maps

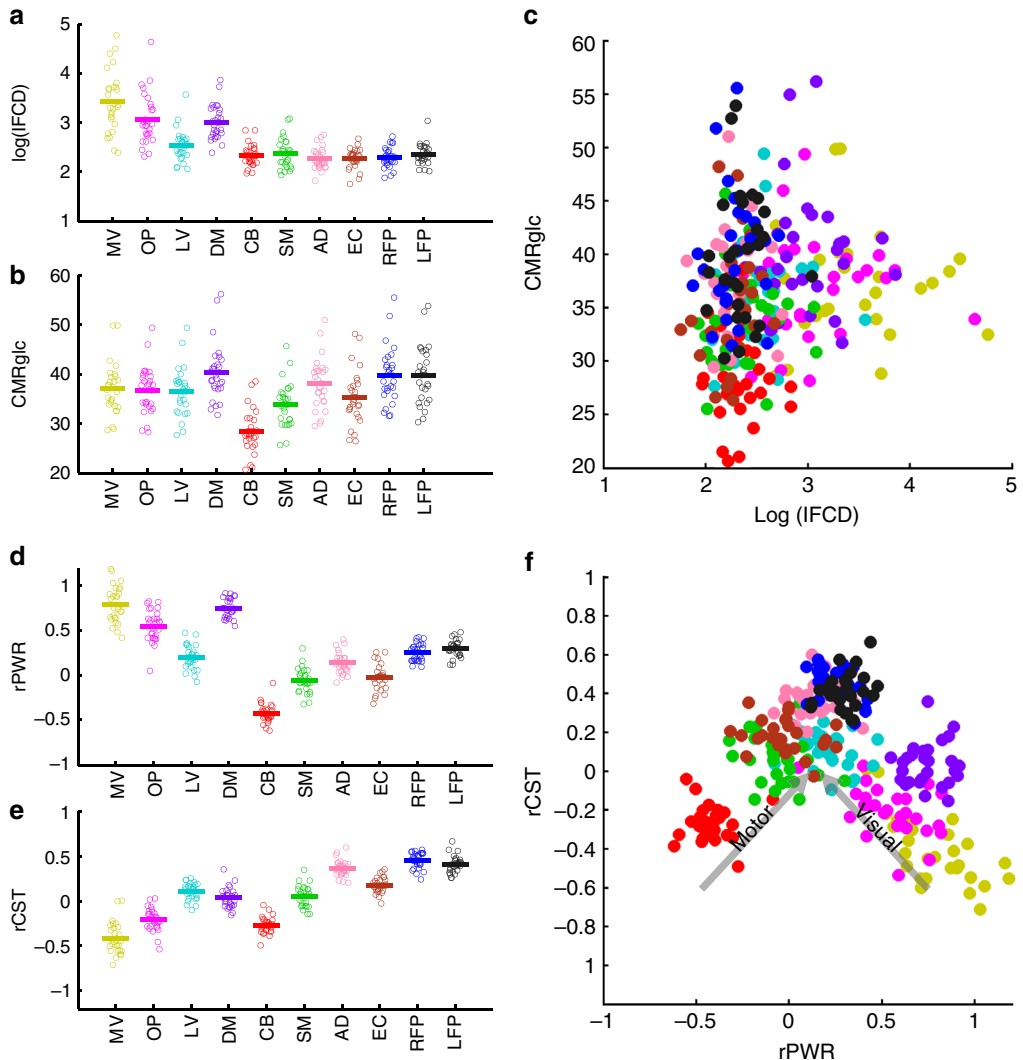

**Fig. 2** Local functional connectivity density (lFCD), cerebral metabolic rate of glucose (CMRglc), relative power (rPWR), and relative cost (rCST) of brain networks and individual differences (cohort-1, $n = 28$). **a**, **b** Within-subject averages (28 circles) and between-subject average (thick line) of lFCD and CMRglc for each of 10 resting-state networks[33] including medial visual (MV), occipital pole (OP), lateral visual (LV), default mode (DM), cerebellum (CB), sensorimotor (SM), auditory (AD), executive control (EC), right and left frontoparietal (RFP and LFP) networks. **c** Within-subject averages of lFCD and CMRglc plotted against each other (each circle represents one participant) for the networks shown in **a**, **b**. The network colors in **c** match those shown in **a**, **b**. **d**, **e** Within-subject averages (28 circles for 28 participants) and between-subject average (thick lines) of rPWR and rCST for each of the 10 resting-state networks shown in parts **a**, **b**. **f** Within-subject averages of rPWR and rCST plotted against each other for the networks shown in **d**, **e**. The network colors in **f** match those shown in **d**, **e**

highlighting alcohol-related changes in lFCD and CMRglc between groups (HD versus LD) and conditions (alcohol (ALC) versus placebo (PLC)). We also computed rPWR and rCST for each participant and condition. Acute alcohol exposure (Fig. 4e) significantly reduced rCST and rPWR in the visual cortex and increased rPWR in the thalamus (Supplementary Tables 13, 14). Chronic alcohol exposure (Fig. 4e) reduced rCST in precuneus (posterior ventral), medial frontal regions, insula, putamen, and CB, and increased rCST in the visual cortex (Supplementary Table 15). Chronic alcohol reduced rPWR in the red nucleus, pons, various thalamic nuclei, and in areas within superior/medial frontal and cingulate gyri and precuneus (not visible in Fig. 4e), but increased relative rPWR in the CB (posterior lobe) and precuneus/cuneus junction (Supplementary Table 16).

**Behavioral association with rPWR and rCST.** In cohort-2, for the subjective measures showing effects of Alcohol, we calculated

a subjective principal component (PC, see Methods) and assessed its association with regional lFCD, CMRglc, rPWR, and rCST in both groups. In ALC, we only found significant negative associations between rPWR in insula, middle, and inferior temporal gyri (in the right hemisphere) and the subjective PC ($p_{FWE} < 0.01$, Supplementary Table 17). While HD performed worse in a range of cognitive tasks, individual differences in these measures in HD (cognitive PC, see Methods) were only significantly associated with rCST in the right inferior parietal lobule ($p_{FWE} < 0.01$, Supplementary Table 18).

**Alcohol and brain energy states.** We characterized the extent to which acute and chronic alcohol altered the brain-wide distribution of rPWR and rCST as well as the distribution of their associations. Despite the zero-mean and unit-variance of whole-brain rPWR and rCST for each participant and condition, ALC relative to PLC significantly increased the skewness (increased

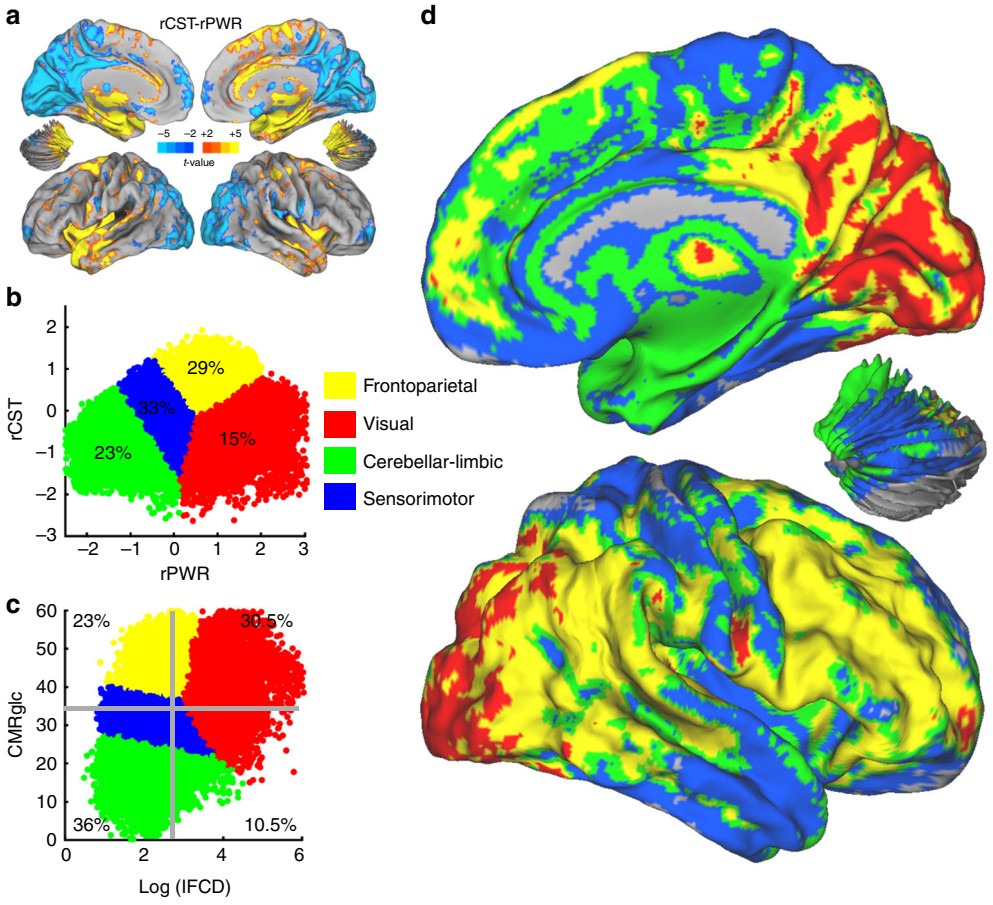

**Fig. 3** Brain segmentation using relative power (rPWR) and relative cost (rCST) (cohort-1, $n = 28$). **a** rCST versus rPWR contrast showing precuneus and visual cortices with significantly lower rCST than rPWR, whereas superior frontal and insular cortices showed higher rCST than rPWR. For highlighting the extreme differences, we used a conservative threshold of $p_{FWE} < 0.01$ at the voxel level ($|t| > 6$). At the conventional threshold ($p_{FWE} < 0.01$, cluster-size corrected), majority of the cortex (54%) showed differences in rCST versus rPWR (see Supplementary Table 3). **b** Scatter plot of rPWR versus rCST for all the brain voxels, showing the four $k$-means clusters with different colors. Numbers represent percentage of voxels falling within clusters. **c** Same clusters projected into the space of local functional connectivity density (lFCD) and cerebral metabolic rate of glucose (CMRglc). The thick gray lines mark the average of log(lFCD) and average of CMRglc with the number in each quadrant showing percentage of voxels falling within that quadrant. These quadrants are defined in Fig. 1e. It is important to note that having a higher percentage of voxels associated with the high- and low-rPWR quadrants is consistent with log(lFCD) and CMRglc being positively correlated (see Fig. 1b). **d** The four $k$-means clusters in **b** on a surface rendering of the brain

bias) of rPWR toward positive values ($F(1, 38) = 7.41$, $p = 0.01$), and decreased the kurtosis (increased uniformity) of rCST ($F(1, 38) = 7.33$, $p = 0.01$) across subjects. We also studied how acute and chronic alcohol affects the distribution of voxels along the four major energy states of rPWR and rCST that were defined relative to the whole-brain averages of rPWR and rCST (Fig. 4g). Acute alcohol reduced the number of (gray matter) voxels associated with lower rPWR and lower rCST state in both groups ($F(1, 24) = 7.81$, $p = 0.01$). While the opposite pattern was seen in the high-rPWR, high-rCST state, this effect was not significant ($F(1, 24) = 3.14$, $p = 0.089$) and was predominantly driven by changes in the skewness of rPWR. LD had higher number of voxels associated with high-rPWR, low-rCST state than HD ($F(1, 24) = 15.44$, $p < 0.001$), but there was a trending interaction with Alcohol factor ($F(1, 24) = 4.01$, $p = 0.057$) (Fig. 4g).

### Discussion
Here we used CMRglc (indexing brain's main energy supply) and lFCD (indexing aspects of neuroglial activity) and defined novel measures of rPWR (extent of concurrent CMRglc and lFCD) and rCST (the extent to which CMRglc leads lFCD) to study the variations in the coupling between CMRglc-lFCD across brain regions. We showed that brain regions can be classified into

major segments with distinct rPWR and rCST characteristics (Fig. 3d) and that resting-state networks differ in their rPWR and rCST (Fig. 2d, e). Specifically, the MV and default mode networks had the highest rPWR, whereas the CB network had the lowest rPWR. Frontoparietal networks had the highest rCST, whereas the MV and CB networks had the lowest rCST. When considering individual differences, we found that brain networks were more segregated based on rPWR and rCST (Fig. 2f) than based on lFCD and CMRglc (Fig. 2c), further supporting the relevance of our approach for studying brain functional specialization. Across cortical regions, rPWR and rCST were not consistently associated with cortical thickness (Supplementary Table 9) and cortical distance (Supplementary Table 10) as were lFCD and CMRglu, suggesting that each of rPWR and rCST measures is differently related to anatomical characteristics of different brain regions. Additionally, we found that rPWR and rCST indices were distinctly sensitive to the effects of acute and chronic alcohol exposure on the brain and behavior.

There were notable differences in regional and network-level rCST (Figs. 1h, 2e). Higher rCST could be attributed to the use of less efficient (but fast) glucose metabolic pathways (e.g., aerobic glycolysis)[16,20], or even a higher glia-to-neuron ratio in the neo cortex[34]. In fact, our estimate of regional rCST (Fig. 1h) had good

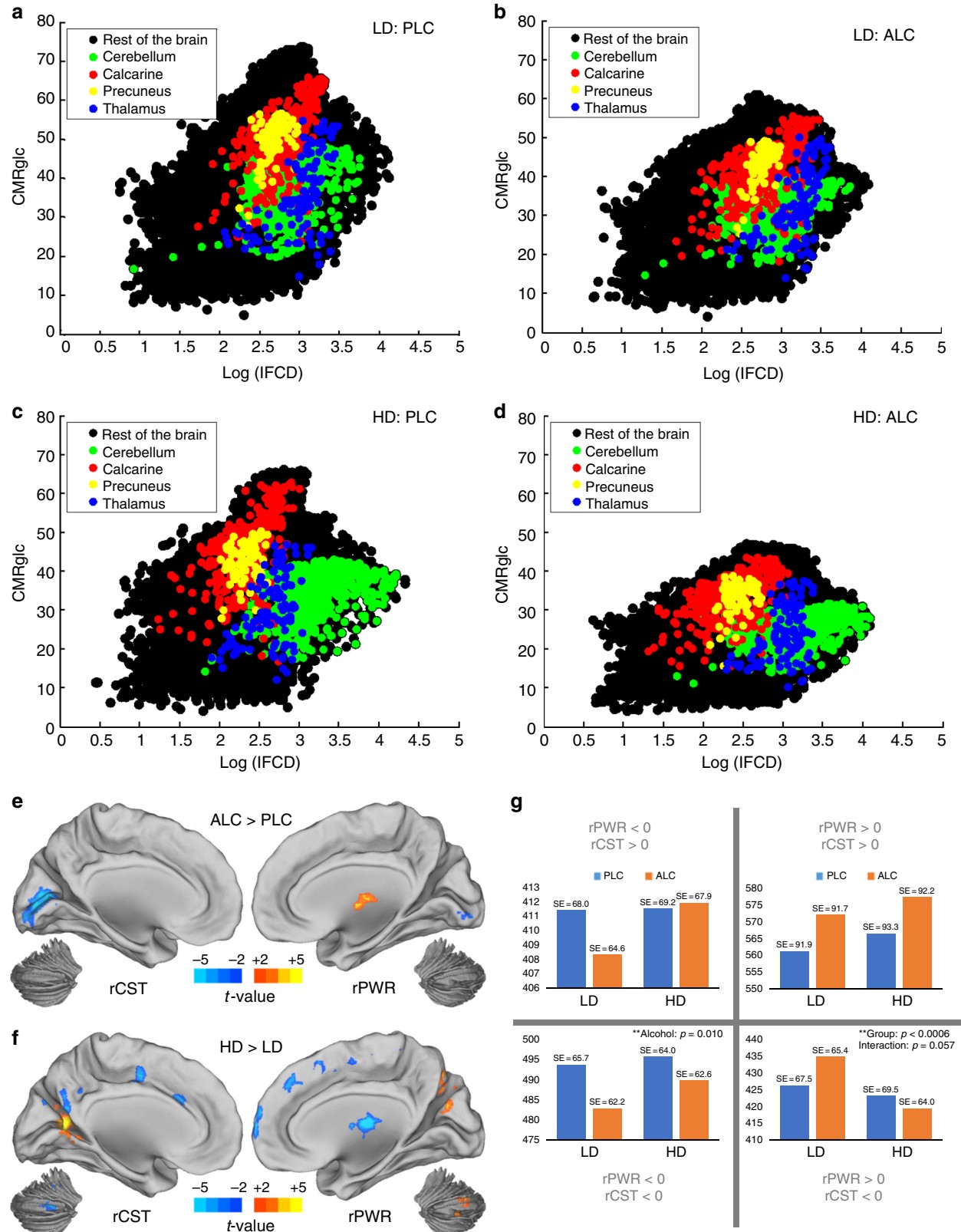

correspondence with previously reported distribution of aerobic glycolysis (accounting for about 10% of glucose metabolized by the adult brain)[16], which highlights dorsal–medial frontal, precuneus, posterior cingulate, and lateral–frontal regions (see Supplementary Table 2). Many instances in the literature attribute higher energetic cost to high regional glucose metabolism[34,35].

Here we provide an approach to quantify energetic cost based on regional glucose metabolism while accounting for underlying activity relative to the rest of the brain (rCST), and expand on prior research that has assessed metabolic cost at the neuronal level[36]. Lower rCST in regions such as the CB (Fig. 1h and Supplementary Table 2) could be attributed to a higher regional

**Fig. 4** Effects of acute and chronic alcohol (cohort-2, $n = 40$). **a–d** COMET (connectivity metabolism) map highlighting the relative distribution of four regions of interest[32] for light drinkers (LDs, $n = 24$) under placebo (PLC) condition (**a**) or alcohol (ALC) condition (**b**), and for heavy drinkers (HDs, $n = 16$) under PLC (**c**) or ALC (**d**). **e** ALC versus PLC contrast for relative cost (rCST, left) and relative power (rPWR, right). **f** HD versus LD contrast for rCST (left) and rPWR (right) indices ($p_{FWE} < 0.01$, cluster-size corrected). No significant interaction effect was found. **g** Statistical analysis of changes in the occupancy of different energy states because of acute (PLC versus ALC) or chronic (LD versus HD) alcohol exposure. Repeated-measures analysis of variance was performed to study Group and Alcohol effects on the 25 frequency bins of each quadrant of rPWR and rCST. Note that average rPWR and rCST is always zero. The bars show average of 25 bins within each quadrant normalized by the number of subjects for each group (see Supplementary Fig. 10 for more details). ** indicates correction for multiple comparisons. SE: Standard Error

proportion of oxidative phosphorylation to aerobic glycolysis[16] or use of energy sources other than glucose such as ketone bodies. Notably, the CB shows the highest levels of acetate metabolism (when plasma acetate levels are increased)[30] and also has the lowest glia-to-neuron ratio, both of which could contribute to its low glucose rCST[37]. Among the resting-state networks, the MV and default mode networks showed the highest rPWR (Fig. 2d). While both networks are among the most active resting-state networks (e.g., as found by independent component analysis)[38], higher rPWR in these regions supported that these networks are also metabolically demanding. We found that all the tested resting networks[33] could be differentiated based on either rPWR or rCST measures (Fig. 2d–e; $p_{FWE} < 0.05$, Sidak). This suggests that functional resting-state networks have distinct energetic signatures that are, in principle, consistent with their functional specialization.

The contrast analysis indicated that most of the cortex had different rCST compared to rPWR (approximately 54% of gray matter voxels, $p_{FWE} < 0.01$, cluster-size corrected). However, the most pronounced differences were in the visual cortices with higher rPWR than rCST and in the limbic and temporal regions with higher rCST than rPWR (see Fig. 3a and Supplementary Table 3). There were also notable differences in rPWR and in rCST between regions (Fig. 2, Supplementary Table 8). Significant differences in regional rPWR indicated that brain regions vary in the level of concurrent activity and metabolism at rest (Fig. 1g, Supplementary Fig. 3). There is evidence that glutamatergic signaling is a significant contributor to both neuronal firing and regional glucose metabolism at rest[39,40], thus we postulate that variations in resting glutamate release could contribute to differences in rPWR among brain regions. The data-driven $k$-means clustering approach showed that the brain regions could be divided into major segments based on their rPWR and rCST. The frontoparietal segment (yellow; Fig. 3b, d) is composed of regions primarily associated with higher-order cognitive abilities (e.g., fluid intelligence)[41]. These regions are metabolically costly, which according to their functional needs may involve the use of faster (but inefficient) metabolic pathways such as aerobic glycolysis or might reflect greater engagement (or density) of supporting glial cells. It has also been postulated that aerobic glycolysis plays an important role in synaptic plasticity and remodeling[20]. For example, the hippocampal formation is among the regions with the highest synaptic plasticity[42]. While the hippocampus had significantly lower rCST and rPWR relative to the rest of the brain (Supplementary Tables 1, 2), it had higher rCST than rPWR (Fig. 3a, Supplementary Table 3), which is consistent with the energetic needs of the hippocampus to support high rates of neurogenesis and synaptic plasticity[42,43].

Despite the growing interest in multimodal brain segmentation in health[44] and disease[45], most approaches have focused on identifying regional boundaries using structural similarities and anatomical and functional connectivity patterns[46]. From a data-driven perspective, here we expanded on prior studies by incorporating information from brain energetics and classified the brain into

major segments based on rPWR and rCST. We show that areas related to visual processing had higher rPWR but lower rCST (red segment; Fig. 3b, d). This observation is consistent with the visual system being highly active, but also relying on efficient oxidative metabolism resulting in a relatively lower rCST[47]. In addition, our data showed that the MV, occipital pole, lateral–visual networks show increasingly higher rCST and decreasingly lower rPWR, respectively (Fig. 2d–f, $p_{FWE} < 0.05$, Sidak). Prior work has suggested MV, occipital pole, and lateral–visual networks are associated with increasingly higher-order cognitive functions, respectively[33], which supports that higher-order functions have higher rCST. Similarly, increases in rCST were observed from the CB, to sensorimotor, and executive control networks (Fig. 2d–f, $p_{FWE} < 0.05$, Sidak), which was consistent with changes from green (cerebellar-limbic) to blue (sensorimotor) segments in Fig. 3b–d. Our data suggested that the evolution of networks toward higher-order functions along a "motor" direction (from CB to sensorimotor and executive control) or along a "visual" direction (from MV to occipital pole and lateral–visual) is associated with consistent increases in rCST (Fig. 2f).

It is worth noting that rPWR and rCST values are relative to the rest of the brain with the assumption that CMRglc indexes glucose energy supply (among other energy substrates) and that synchronous regional fluctuations in the BOLD signal are proportional to local activity, each capturing a fraction of energy expenditure and neuroglial activity, respectively. Using CBF (PWI) and fALFF as alternative indices of metabolic supply and neuronal activity resulted in rPWR and rCST that had a strong agreement with those obtained with CMRglc and lFCD (Supplementary Results, Supplementary Figs. 6, 7), further supporting the generalizability of these metrics (see also Supplementary Discussion on lFCD and fALFF). However, we cannot rule out that inherent limitations in PET and MRI imaging (e.g., spatial heterogeneity in tSNR) could to some extent affect regional differences in rPWR and rCST. We observed that rPWR and rCST do not appear to amplify the effects of tSNR in fMRI and FDG-PET modalities (Supplementary Results, Supplementary Tables 11, 12, and Supplementary Fig. 8). Visual networks showed significant differences in rCST and in rPWR (Fig. 2d, e), yet they had mid-to-high range tSNR in both modalities (Supplementary Tables 11, 12), suggesting that differences in rPWR and rCST in visual regions are not primarily related to measurement noise. Morphometric properties of brain regions are expected to be consistent with their functional and energetic needs[48]. We found that average cortical thickness and cortical distance (see Methods) in several regions were positively associated with rCST and rPWR ($p < 0.05$, Bonferroni, Supplementary Tables 9, 10). It could be expected that higher cortical distance (or thickness) is consistent with greater activity and energetic needs, particularly for regions with significant remote connectivity. Interestingly, cortical distance and rCST were associated in entorhinal, lateral orbitofrontal, and inferior temporal cortices, which are among regions implicated in Alzheimer's disease[49,50]. Nevertheless, the relevance of these morphological findings

remains to be determined. Our work is limited to assessing activity demand and metabolic supply from a static perspective, but future work should involve studying dynamic changes in rPWR and rCST due to physiological[51] or circadian[52] interactions with brain function, using concurrent imaging of neuronal activity (e.g., measures of glutamatergic function or lFCD) and cerebral metabolic supply (e.g., CMRglc or CBF[53,54]).

We found significant regional changes and differences in rPWR and in rCST due to acute and chronic alcohol exposure (Fig. 4e–f). We interpreted less regional rPWR in HD (e.g., thalamus and medial frontal regions) to reflect toxic effects of long-term exposure to alcohol on neuronal and glial cells, leading to reduced activity and glucose metabolism. The decreases in rPWR with acute alcohol, which were predominantly observed in the visual cortex, are likely to reflect disruption of visual processing associated with alcohol intoxication[55,56] (Supplementary Tables 14, 16). Increases in rPWR may indicate compensatory phenomena with elevated activity/metabolism[32], for example, during intoxication in the thalamus, or as a result of chronic alcohol use in the CB and (posterior medial segment of) precuneus (Supplementary Tables 14, 16). The visual cortex had the most significant decreases in rCST during alcohol intoxication, which is consistent with prior studies showing that this brain region had the largest decrements in glucose metabolism and one of the largest increases in acetate metabolism during intoxication[9,30]. Less rCST in cortical and cerebellar regions in HD may indicate prolonged dependence on ketone bodies[30] or glial cell loss[57], whereas higher rCST of visual areas in HD than LD could indicate an increased rate of aerobic glycolysis due to repair mechanisms[58] (Supplementary Tables 13, 15). However, future research is needed to confirm these speculations. Relative to lFCD, CMRglc, and rCST, less rPWR was associated with greater subjective experience of alcohol in the insula (Supplementary Table 17), suggesting that concurrent changes in (BOLD) activity and glucose metabolism in this limbic region provide relevant markers of subjective alcohol experience[59]. We also found evidence that rCST (relative to lFCD, CMRglc, and rPWR) in the inferior parietal lobule was positively associated with performance in cognitive tasks in HDs (Supplementary Table 18). While activity in parietal regions have been previously related to intelligence[60], our findings suggest that energetic needs (i.e., rCST) in parietal areas contribute to individual differences in cognitive performance in HD.

Acute alcohol altered global distribution characteristics of rPWR and rCST by increasing its skewness towards positive values, where a larger number of regions showed concurrently high metabolism and high activity levels. This observation is consistent with an increased coupling between glucose metabolism and synchronous local activity for high functioning regions after intoxication. It has been shown that glial cells, but not neurons, are primarily able to metabolize acetate[61], thus reliance of glial cells on acetate metabolism after intoxication could result in a stronger coupling between neuronal activity and glucose metabolism. Decreased kurtosis of rCST (longer distribution tails) from PLC to ALC indicated that there were increases in the number of brain voxels with higher rCST and lower rCST. This observation is consistent with findings that regional decreases in glucose metabolism during intoxication are not homogeneous and reflect, in part, low to high reliance on acetate metabolism across regions[30]. Chronic alcohol exposure also altered the relative distribution of rCST and rPWR (Fig. 3g). We found that LD had more voxels associated with the higher rPWR and lower rCST energy state (higher efficiency) than HD, while the opposite pattern was seen in the higher rCST and lower rPWR state (Fig. 4g). Despite global decreases in glucose metabolism in HD compared to LD (Fig. 4a–d), our data indicated that the HD brain is shifted

toward less efficient energetic states (indicated by rPWR and rCST), but future studies are needed to investigate the mechanisms that could contribute to this relatively inefficient state of energy utilization (including the role of inflammation[62]).

The associations between activity demand and metabolic supply in the brain are important for studying brain function[20,22,23] and diseases[24]. In principle, energy demand and supply in the brain are matched (with certain exceptions such as heat[63] or lactate[64] production). However, our measures of activity demand (i.e., lFCD) and metabolic supply (i.e., CMRglc) only capture specific aspects of demand and supply processes in the brain, making their mismatch not only possible but also informative. Conditions such as exposure to alcohol[30] or sleep[21] are known to impact brain glucose metabolism due to the use of alternative sources of energy (e.g., acetate) or changes in active glucose metabolic pathways (aerobic glycolysis versus oxidative metabolism). However, without accounting for underlying brain activity, changes in glucose metabolism are hard to interpret. To quantify the relative changes in the association between two modalities, here we proposed two novel metrics of rPWR and rCST that are unit-free and are generalizable to measures of brain activity and energy supply. These metrics quantify how well two modalities that measure aspects of activity demand (such as lFCD) and aspects metabolic supply (such as CMRglc) are concurrently high or low (i.e., rPWR) and to what extent one exceeds the other (i.e., rCST) in a specific region relative to the rest of the brain. From another perspective, rPWR could be thought of as an index of concurrent *intensity* of the two modalities, while rCST is an index of mismatch between the two modalities. Since rPWR and rCST are relative, they are not sensitive to global changes in any of the two modalities (e.g., levels of glucose metabolism as in acute or chronic alcohol exposure, Fig. 4), and could be useful to map and track the associations between specific markers of activity demand and metabolic supply under different physiological conditions (e.g., sleep), pharmacological states (e.g., alcohol intoxication), or disease stages (e.g., Alzheimer's disease). Analysis of individual differences showed a clear segregation of different brain networks based on rPWR and rCST (Fig. 2f) relative to lFCD and CRMglc (Fig. 2c). We found that rPWR and rCST are robust and generalizable to other measures of activity demand and metabolic supply (e.g., fALFF and CBF, Supplementary Figs. 6, 7) and do not appear to amplify the effects of measurement noise on lFCD and CMRglc (Supplementary Tables 11, 12). Analysis of the whole-brain distribution of rPWR and rCST characterized alterations in the lFCD-CMRglc coupling during acute alcohol exposure and showed that prolonged alcohol use may shift the brain toward less efficient energetic states. More importantly, we found that rPWR and rCST (relative to lFCD and CMRglc) were significantly and distinctly related to behavioral effects of acute and chronic alcohol exposure, further supporting their utility for capturing meaningful (and possibly unique) aspects of brain function. Thus, we propose rPWR and rCST as new multimodal metrics to study the energetic economy of brain networks[65] throughout the lifespan and to monitor the effects of drugs and diseases on the human brain.

## Methods

**Participants**. Here we report data in two independent cohorts ($n = 68$ total). In cohort-1 (NM; $n = 28$; age = $36 \pm 12$ years; 17 males), we collected high-resolution resting-state fMRI and CBF (PWI) (two sessions) and FDG-PET (one session) (raw fMRI and FDG-PET data partly reported[66]). Cohort-1 data were collected in healthy participants at the National Institutes of Health (NIH) with no intervention. Cohort-1 participants provided written informed consent to participate in the study, which was approved by the Institutional Review Board at the NIH (Combined Neurosciences White Panel). In cohort-2 ($n = 40$), participants underwent resting-state fMRI (two sessions)[32] and FDG-PET imaging (two sessions)[9] at the Brookhaven National Laboratory. Signed informed consents were obtained from

the participants as approved by the Committee on Research in Human Subjects at Stony Brook University. Cohort-2 consisted of HDs ($n = 16$; age $= 34.6 \pm 9.7$ years; 16 males) and LDs ($n = 24$; age $= 32.5 \pm 6.4$ years; 12 males). Each participant was tested twice under resting state: once during PLC (one fMRI session and one FDG-PET session) and once during ALC (one fMRI session and one FDG-PET session). In brief, HD consumed five or more drinks in each day on three or more occasions per week and reported last use of alcohol within 3 days of the imaging session. The LD had prior experience with alcohol, but at most consumed one drink in any given day. Exclusion criteria were: (1) urine positive for psychotropic drugs; (2) history of alcohol or drug use disorders (except nicotine use for all participants and alcohol use disorder for HD); (3) present or past history of neurological or psychiatric disorder; (4) use of psychoactive medications in the past month (i.e., opiate analgesics, stimulants, sedatives); (5) use of prescription (non-psychiatric) medication(s), that is, antihistamines; (6) medical conditions that may alter cerebral function; (7) cardiovascular and metabolic diseases; and (8) head trauma with loss of consciousness of more than 30 min.

**Cohort-1 data acquisition**. Each participant underwent MRI on two separate days (with an average of 20.4 days and a median of 8 days apart) and one FDG-PET scan that was acquired either on the first or second MRI day. The MRI was done on a 3.0 T Magnetom Prisma scanner (Siemens Medical Solutions, Erlangen, Germany) using a 20-channel head coil. On each MRI day, resting-state fMRI was collected for 15 min using a single-shot gradient echo-planar imaging sequence (echo time (TE) $= 30$ ms, repetition time (TR) $= 1.5$ s) with 3-mm in-plane resolution and 4-mm slice thickness (no gap). The fMRI scan was performed while participants relaxed with eyes open during the scan (no fixation cross). On each MRI day, we also collected pulsed arterial spin labeling data using a 3D gradient and spin-echo sequence (TR/TE $= 2300/16.18$ ms) with 3-mm in-plane resolution, 3-mm slice thickness (48 slices), bolus duration $= 700$ ms, inversion time $= 1600$ ms, and turbo factor $= 18$ (total acquisition time 4:59 min). Siemens in-line processing provided a summarized PWI from four pairs of tagged and untagged images, which was used as a relative proxy of CBF. For each participant, the summarized CBF (PWI) map was averaged across the two scan days, except for one subject with only one available CBF (PWI) scan day. T1-weighted 3D MPRAGE (TR/TE $= 2200/4.25$ ms, 1-mm isotropic resolution) and T2-weighted spin-echo multi-slice (TR/TE $= 8000/72$ ms, 1.1 mm in-plane resolution, 1.7 mm slice thickness, 94 slices) pulse sequences were used to acquire anatomical brain images. In the paper, glucose metabolic supply refers to energy utilization (not to be confused with glucose delivery) that was indexed by CMRglc estimated by FDG-PET scans. FDG-PET scan was performed using a high-resolution research tomography (Siemens Medical Solutions, Knoxville, TN, USA) with ~2.5 mm camera resolution. Two venous catheters were placed, one to measure the concentration of radioactivity from arterialized venous blood and the other one for radiotracer injection. Prior to tracer injection, a transmission scan was obtained using cesium-137 to correct for attenuation. [18]FDG (8 mCi) was injected intravenously over a period of approximately 1 min. PET emission scans were obtained in list mode (one image every 10 s) starting immediately after [18]FDG injection for 75 min. During the PET imaging procedure, the subjects rested quietly under dim illumination and minimal acoustic noise. To ensure that subjects did not fall asleep, they were monitored throughout the procedure and were asked to keep their eyes open (no fixation cross). A cap with small light reflectors was used to monitor head movement and to minimize motion-related image blurring. A summary image was obtained between 35 and 75 min (reconstructed with a 1.23-mm isotropic voxel size). See Supplementary Methods for details on data preprocessing.

**Cohort-2 data acquisition**. Each participant was tested twice (on separate days; maximum 3 days apart): once during ALC and once during PLC in a random order[9]. Participants drank alcohol (0.75 g/kg mixed in a caffeine-free diet soda) or placebo (caffeine-free diet soda) within a 20-min period under single-blind conditions. We used a specialized drinking container with an alcohol-containing lid that provided the smell of alcohol and delivered the same volume of liquid in both conditions. Participants were injected with 4–6 mCi of [18]FDG about 40 min after drinking onset. The [18]FDG uptake period lasted 35 min during which participants sat on a chair in a dimly lit room with minimum acoustic noise, were periodically assessed for the effects of alcohol or placebo, and were continuously monitored to ensure that they kept their eyes open and did not fall asleep. PET scans were done using a Siemens ECAT EXACT HR+ tomograph for 20 min, where participants were positioned using an individually customized head holder. Transmission scans were obtained using germanium-68 to correct for attenuation. FDG-PET images were reconstructed using filtered back projection (Hann filter with a 4.9 mm full-width at half-maximum kernel) that were used for CMRglc estimation. The MRI scans were acquired between 90 and 120 min of alcohol or placebo administration following the FDG-PET scan. At the beginning of the MRI scanning session in ALC, the average blood alcohol concentration was 0.62 mg/ml (SD $= 0.27$ mg/ml). Resting-state fMRI data were collected in a 4.0 T Varian/Siemens MRI scanner using a T2*-weighted single-shot gradient-echo planar imaging sequence (TE $= 20$ ms, TR $= 1.6$ s) with 3.1-mm in-plane resolution and 4-mm slice thickness (1-mm slice gap). Participants were instructed to remain silent, motionless, and awake with their eyes open during the 5-min resting-state fMRI (no fixation cross). See Supplementary Methods for details on data preprocessing.

**Behavioral measures**. In cohort-2, behavioral measures were obtained in 23 LD participants (out of 24) and 15 HD participants (out of 16)[9,32]. We found significant effect of ALC for five self-reported measures of feeling sedated, dizzy, high, pleasant, and intoxication[32]. For summarizing subjective effects of ALC in this study, we computed the first principal component of these five measures (subjective PC accounted for 68% of variance), in ALC across both groups. We also found HD had lower cognitive performance in five tasks: Stroop (neutral, congruent, and incongruent), Symbol Digit Modalities Test, and Word Association tasks[32]. To summarize individual differences in cognitive performance in HD, we computed the first principal component of these five measures obtained during PLC in HD (cognitive PC accounted for 50% of variance).

**lFCD estimation**. Prior work has shown a relatively high spatial spread of neuronal activity relative to the stimulation locus[67,68]. This evidence supports that the spatial spread of synchronous BOLD signal is proportional to regional neuroglial activity. Thus, we used the measure of lFCD, quantifying extent of spatial synchrony in slow BOLD fluctuations (0.01–0.1 Hz), to index spontaneous brain activity demand. For both cohorts, fMRI data was band-pass filtered (0.01–0.10 Hz) to remove magnetic field drifts and to minimize the effects of physiologic noise on high-frequency components. For excessive motion, fMRI time points that were severely affected by motion were removed using a "scrubbing" approach[69] with a root mean square signal change (DVARs) threshold of 0.5% and a framewise displacement threshold of 0.5 mm. Remaining motion effects on fMRI time series were regressed out using estimates of motion parameters. Pearson's correlation was calculated to assess the strength of functional connectivity, $C_{ij}$, between voxels $i$ and $j$. We define local functional connectivity graph $G = (V, E)$, such that brain voxels are its vertices $V = \{v_1, \ldots, v_n\}$, and there is an edge between $v_i$ and $v_j$ if $C_{ij}$ is larger than 0.6[26,70] and if $v_j$ is part of a cluster of voxels that are spatially connected to $v_i$ (using a surface or edge criterion). lFCD (or local functional connectivity degree) was defined as the number of edges associated with $v_i$. Because degree-related measures (such as lFCD) follow an exponential distribution, we used log(lFCD) in all analyses with a semi-normal distribution (Supplementary Fig. 2) to characterize brain activity and its associations with brain glucose metabolism (indexed by CMRglc). For cohort-1, the average log(lFCD) from the two fMRI sessions were used in the analyses to improve the SNR of brain activity measures. In cohort-1, we tested the sensitivity of rPWR and rCST to lFCD threshold by also using an alternative threshold of $C_{ij} = 0.4$ and compared the results to those obtained with lFCD threshold of $C_{ij} = 0.6$. The effects of unwanted fluctuations within the white matter and cerebrospinal fluid were excluded from the analysis by using a mask of gray matter for calculations of indices.

**fALFF estimation**. As an alternative measure of brain activity at rest (cohort-1), we computed fALFF using 3dRSFC in AFNI[71], while regressing out motion regressors, to quantify the relative contribution of 0.01–0.1 Hz frequency range to the fMRI spectra. For cohort-1, the average fALFF from the two fMRI sessions were used in the analyses to improve SNR of brain activity measures.

**rPWR and rCST estimation**. Voxelwise rPWR and rCST were computed by a $\pi/4$ rad (45°) counterclockwise rotation of (mean and variance normalized) log(lFCD) and CMRglc axes, respectively. Specifically, in a two-dimensional polar coordinate system of standardized (whole-brain mean $= 0$, variance $= 1$) log(lFCD), $z$(log(lFCD)), plotted against standardized (whole-brain mean $= 0$, variance $= 1$) CMRglc, $z$(CMRglc), we define:

$$\text{rPWR} = R \times \cos\left(\theta - \frac{\pi}{4}\right) \qquad (1)$$

and

$$\text{rCST} = R \times \sin\left(\theta - \frac{\pi}{4}\right), \qquad (2)$$

where $R$ and $\theta$ are radius and angle (in radians) of each brain voxel in a polar coordinate system of $z$(log(lFCD)) and $z$(CMRglc). Alternatively, in a Cartesian coordinate system,

$$\begin{bmatrix} \text{rPWR} \\ \text{rCST} \end{bmatrix} = \begin{bmatrix} \cos\frac{\pi}{4} & \sin\frac{\pi}{4} \\ -\sin\frac{\pi}{4} & \cos\frac{\pi}{4} \end{bmatrix} \begin{bmatrix} z(\log(\text{lFCD})) \\ z(\text{CMRglc}) \end{bmatrix}. \qquad (3)$$

This 45° counterclockwise rotation is equivalent to performing a principal component analysis on two positively correlated variables that are, mean and variance normalized (resulting in an equal contribution of imaging parameters into rPWR and rCST). It is important to note that rPWR and rCST variables are orthogonal, thus it is possible for brain regions to have high (or low) rPWR and rCST at the same time, while other regions may be high in rCST and low in rPWR or vice versa. While rPWR captures most of the common variance, rCST captures the reminder of the common variance between $z$(log(lFCD)) and $z$(CMRglc). This is evident in Supplementary Fig. 3 histograms, showing higher variability across regions in rPWR (capturing most of the variance) relative to rCST. rPWR should not be confused with other uses of term "power" in the literature such as power in electrical circuits (though they bare some similarity), statistical power, and power in time series. Voxelwise rPWR and rCST

were estimated for each participant in cohort-1 and each condition in cohort-2 (e.g., PLC and ALC, see Supplementary Fig. 9).

**Network segregation index**. NSI was defined to index how well a brain network (or a region) is segregated from other brain networks (or regions), when individual participant data for all the networks are plotted in a two-dimensional space (e.g., Fig. 2c versus Fig. 2f). For this purpose, three parameters were calculated. For a given network ($N_i$), center of mass ($CM_i$) coordinates were calculating by averaging of each dimension across participants. For $N_i$, average within network distance ($AWND_i$) was calculated by averaging the Euclidian distance of participants data points from the $CM_i$. For $N_i$, average between network distance ($ABND_i$) was calculated by averaging the Euclidian distance from $CM_i$ to the CMs of the rest of the networks. Finally, NSI was defined as following, which is expected to have a $F$-distribution:

$$\text{NSI}_i = \frac{\text{ABND}_i}{\text{AWND}_i}. \tag{4}$$

***k*-means clustering**. For classifying brain regions based on the associated rPWR and rCST indices, a $k$-means clustering approach[72] was used in MATLAB (The MathWorks, Natick, MA, USA). In brief, $k$ centroids (cluster centers) with $l$-dimensions ($l = 2$ here for rPWR and rCST) were first randomly selected ($k$ is a pre-specified number). Each of the $n$ observations ($n$ = number of brain voxels) was assigned to the closest centroid. The algorithm iteratively updates the centroids' location so that within-cluster distances to the centroid, decreases at each iteration. Specifically, an updated cluster centroid was computed by averaging the observations in each cluster. This process continued until the maximum number of specified iterations was reached ($i = 1000$) or there was no change in cluster assignments. Clustering was performed on the group-level average maps of rPWR and rCST. Clusters did not change when the $k$-means clustering process was repeated ($n = 100$).

**Statistical parametric mapping**. SPM8 was used to perform voxelwise comparisons on rPWR and rCST indices. For the cohort-1, one-sample $t$ tests were used to identify regions with high and low rPWR and rCST relative to the rest of the brain. A paired $t$ test was used to identify regions that are significantly different in rCST and rPWR. For cohort-2, a flexible factorial analysis was used to model between-subject factor of Group (LD versus HD) and within-subject factor of Alcohol (PLC versus ALC), where gender and smoking status were entered as covariates to control for group differences in these variables. All effects were corrected for multiple comparisons using cluster size correction approach ($p_{FWE} < 0.01$) and a cluster-forming threshold of $p < 0.005$. When indicated, we used a more stringent threshold (corrected at the voxel level, $p_{FWE} < 0.01$, $|t| > 6$) to further guide summarizing large effects.

**Brain morphometry and temporal SNR**. In cohort-1 ($n = 28$), we used 34 surface-based cortical parcellations in each hemisphere (a total of 68 regions of interests (ROIs)) in FreeSurfer (see Supplementary Methods) to study how brain morphometry and fMRI tSNR and FDG-PET tSNR measures are associated with lFCD and CMRglc as well as rPWR and rCST. FreeSurfer pipeline provides surface-based cortical thickness estimates for each of the 34 ROIs in each hemisphere (lh.aparc.stats and rh.aparc.stats). For each bilateral ROI, we averaged thickness measures between left and right hemispheres. We also computed a measure of cortical distance (in the subject space before MNI normalization), specifically, by computing the average geometrical distance from the center-of-mass of each ROI to center-of-mass of the rest of 67 ROIs. For each bilateral ROI, this measure was averaged between left and right hemispheres. For fMRI, voxelwise tSNR was computed using the mean to standard deviation ratio of the raw fMRI time series (after motion correction and spatial normalization). For FDG-PET, voxelwise tSNR was computed using the mean to standard deviation ratio of the dynamic FDG time series (after motion correction and spatial normalization, acquired between 40 and 55 min, matching the length of fMRI sessions). For each subject, each tSNR measure was averaged within each of the 34 bilateral ROIs.

**Code availability**. The code for estimation of metrics will be made available at https://github.com/eshoko/COMET.

## Data availability

The authors declare that data supporting the study findings are available within the paper and its Supplementary Information files and are available upon request. The cohort-1 dataset collected at the NIH is registered in clinicaltrials.gov under registration code: NCT02193425 as an "early phase 1 trial".

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

## Acknowledgements

This study was supported by NIH/NIAAA intramural research program (Y1AA3009). We are grateful for input from William Dieckmann, Sunny Kim, and Lalith Talagala. We thank Shantalaxmi Thada and Shielah Conant for help with PET data at the NIH. We also thank Samantha Cunningham and Min Guo for the kind support, Lori Talagala, Elizabeth Cabrera, Elsa Lindgren, Gregg Miller, and Emily Skarda for assistance with MRI and PET scanning at the NIH, Ruiliang Wang for assistance with BNI data collection, and Chris Wong and Karen Torres for help with data and protocol management.

## Author contributions

E.S.-K., D.T. and N.D.V. initiated the project. E.S.-K. developed the metrics and performed the analyses. E.S.-K. and N.D.V. wrote the paper. E.S.-K., D.T., C.E.W, and G.-J.W. collected the data. B.A. assisted with segmentation analysis. All authors reviewed or edited the paper.

## Additional information

**Competing interests:** The authors declare no competing interests.

