## [Peer Review File · Nature Communications]

Reviewers' comments:

Reviewer #1 (Remarks to the Author):

The authors have submitted a study which uses PET and fMRI data recorded from human subjects. A “supply” factor is calculated from PET data and a “demand” factor from fMRI. These factors are then transformed with sinusoidal transformation to provide two new factors, “power” (which corresponds roughly to correlation or dependence between supply and demand) and “cost” (which corresponds roughly to anti-correlation or independence between supply and demand). These factors are then statistically analyzed within healthy controls to derive resting state networks. Following this, analysis is done under acute or chronic alcohol exposure to show differences in cost and power. Finally, demand (but not supply, cost or power) is shown to change under visual stimulation.

The study is generally well-written and its use of multimodal data to examine alcoholism is very interesting. Unfortunately, it is problematic in that the terms used are confusing, the derived metrics are poorly justified, and independent verification was generally not done. To better clarify these criticisms:

Confusing terms:

1. Prior studies seem to refer to “supply” as CMRglc and “demand” as IFCD. I would recommend using the original terms. Regarding supply, this study itself acknowledges aerobic glycolysis which (when the “glycolytic index” measure is examined) shows that oxygen metabolism and glucose metabolism can be very different. Regarding demand, some of the authors’ prior work has shown that IFCD does not cover every form of fluctuation measurable with resting state fMRI in its relationship to CMRglc.

2. “Power” already has a commonly used meaning in time series, and as this study uses fMRI time series to calculate IFCD, its use here is confusing. “Cost” has clearer meaning here (than supply, demand, or power) but ignores that glucose may be being used for purposes other than IFCD, causing the change in “cost.”

Poor justification of derived metrics:

3. Prior studies by some of the authors and others well-characterized both similarities and differences between CMRglc and IFCD (and other fMRI measures). What is gained by the transformation from supply/demand to cost/power? Most tests are only done on supply/demand but not cost/power. Do cost/power provide fundamentally different information? They are further derived and thus harder to conceptualize measures, for this reason the authors need to justify what they provide that the less derived measures do not.

4. Why is the transformation sinusoidal instead of a projection or coordinate transformation? This seems to create a gap in the cloud distribution (Figure 1D/F). Is this desirable? Why? Why does it then look like a rotation in 2D space (Figure 2A/B), which hypothetically shouldn't matter for clustering?

5. The major focus of the study is power/cost, so why is the stimulation data for which PET appear to be unavailable used? If power and cost provide information supply and demand do not, then this isn't comparable as power and cost cannot be calculated without supply data.

Independent verification not done:

6. This is a multimodal study and, while prior work has examined the effect of PET and fMRI being recorded through very different methods, this study calculates new measures from them. When new measures are derived, small portions of the variance in source data could be amplified. Thus it is critical to validate that the new measures are not amplifying noise. One area in particular is the visual cortex which appears strongly in this study with cost/power but can often have good signal in PET but poor signal in fMRI. At a minimum, the cost/power calculation and SPM and k-means clustering (same number of clusters) should be done on SNR maps for PET and fMRI to see if the SNR variation between PET and fMRI causes some of the clusters.

7. Cost and power would be more justifiable (versus the existing measures of CMRglc and IFCD) if they were independently verified where existing measures were not. E.g. the authors could use EEG to measure demand and MRS to measure supply in only two widely separated brain regions with high cost/power deviation. There are many other possible ways to test this, but demonstration that new metrics have validity outside of combined PET/fMRI would go a long way to validating them.

8. Voxel-based morphometry (VBM) is commonly used in alcohol use and alcoholism. The relative gray matter/white matter levels are thus of suspicion when comparing PET and fMRI modalities. (The authors limit their analysis to gray matter, but VBM applies within gray matter alone.) Comparing the cost/power significance versus VBM significance would also help validate these

measures as something novel. (Though if VBM differences and cost/power differences are too similar it is unknown whether activity is different due to different anatomy, or if anatomy creates an artifact which alters activity measurement.)

The noted problems are addressable but require a substantial rewrite with substantial new data analysis.

Reviewer #2 (Remarks to the Author):

The authors acquired BOLD-fMRI and FDG-PET data from healthy subjects and devised a novel, integrated measure of power/cost to identify regions with high/low fMRI activity at high levels of energy utilization. They found distinct power-cost distributions across the cortex of healthy subjects with characteristic changes in another cohort of alcohol drinkers.

While I found it highly interesting to gain novel insights into human brain organization from multimodal imaging, I have major concerns with the methodology of defining this novel measure of power/cost and was less convinced by the extended view about brain organization derived from the novel approach.

The measure of IFCD is a tightly defined parameter to capture neuronal activity from the BOLD-fMRI signal. It captures the synchrony of BOLD-signal fluctuations between a specific voxel and voxels in close neighborhood at a particular threshold C which is then log-transformed. In order to relate fMRI-derived neuronal activity to absolute glucose metabolism, I would suggest to offer a broader perspective on BOLD-measures. How do parameters used in this study such as thresholding, log-transformation, pearson-correlation instead of signaling amplitude impact on the relation of IFCD to CMRglc and do other parameters such as structural distance or cortical thickness possibly relate to the regional distribution of power/cost? It is important to note that the novel power/cost measure was only derived from $N=17$ healthy subjects of a total of $N=75$. From those 17 subjects, was the PET data always related to the fMRI from the same day or possibly from the second MRI scan?

The authors report a strong linear correlation between logIFCD and CMRglc (Suppl Fig 1) which I found not interpretable with respect to the raw data presented in Fig 1B of the main manuscript. The heat plot in Fig 1B rather suggests a very confined and heavy aggregation of voxels towards the lower left end of the graph with almost no variance along the logIFCD axis. With respect to the slope in Suppl Fig 1 this linear correlation is hardly justified when plotting onto the heatmap data.

Moreover, I did not understand how the regular/circular distribution of power/cost from the model (Fig 1E) would fit onto the heavy tailed voxel distribution in Fig 1B. It seems that >90% of all voxels would fall into the low power-quadrant making it hard to interpret whole cortex data. Together, this critique is also reflected by the fact that the novel measure of power/cost (Fig 1 G/H) reveals an almost identical spatial distribution as CMRglc (Fig 1C) alone which indeed has strongest variance in Fig 1B. Several voxels seem to have highest relative levels of both power and cost (lateral prefrontal regions). Doesn't that contradict each other?

Overall, I found the research question and concept of integrating both imaging modalities very interesting. However, I don't see the major claims supported by the data and would suggest a more detailed derivation of the complex measures power/cost from the fMRI data.

Reviewer #3 (Remarks to the Author):

In this manuscript, the authors have defined novel measures for characterizing coupling between glucose metabolism (measures with PET imaging) and resting state connectivity (measured with fMRI). They came up with two measures; power (extent of high activity and energy utilization) and cost (the extent to which energy utilization exceeds activity), and showed that different regions of the brain has different ratios of these measures. They also showed that these measures may be altered in heavy alcohol drinkers, and with acute alcohol administration. While this is an interesting concept, questions remain about how robust and reliable these measures are, whether they are highly variable between individuals, or whether they are affected by limitations of the technology itself (e.g. scanner-related issues showing stronger signal to noise ratio (SNR) in some regions compared to others).

It strikes me that areas of high power may in part be conflated with areas of the brain that are more easily imaged by fMRI (e.g. visual areas and default-mode), whereas areas of low power may be harder to image due to signal dropout. Have the authors considered this, or run any analyses of whether SNR maps were significantly correlated with power/cost in these areas?

I am not sure of the purpose of the the first 2 cohorts (NIH1, n = 28, and NIH, n = 7). Did the authors use these two cohorts to assess test-retest reliability? I did not see any data indicating that cohort 2

was used to verify the results of cohort 1. Were the data combined from these two cohorts? Please clarify. Similarly, how variable are these measures (cost/power) between individuals?

The alcohol cohort is interesting, yet needs more explanation. Were the power/cost measures sensitive to the level of alcohol drinking, or the level of intoxication, in participants? Do the authors have any speculations about how these measures would lead to the functional outcomes experienced by long-term, heavy drinkers? Would the authors expect an association between power/cost measures and use of alternative sources of energy during intoxication?

It would be interesting to show side-by-side maps of (1) glucose metabolism (PET), (2) resting-state activity (fMRI) and (3) cost/power maps in order to directly visualize how these measures relate to one another. Showing these in the control cohorts and the alcohol cohort would help the reader visualize the potential utility of these measures for clinical purposes.

Reviewers' comments:

Reviewer #1 (Remarks to the Author):

The authors have submitted a study which uses PET and fMRI data recorded from human subjects. A “supply” factor is calculated from PET data and a “demand” factor from fMRI. These factors are then transformed with sinusoidal transformation to provide two new factors, “power” (which corresponds roughly to correlation or dependence between supply and demand) and “cost” (which corresponds roughly to anti-correlation or independence between supply and demand). These factors are then statistically analyzed within healthy controls to derive resting state networks. Following this, analysis is done under acute or chronic alcohol exposure to show differences in cost and power. Finally, demand (but not supply, cost or power) is shown to change under visual stimulation.

The study is generally well-written and its use of multimodal data to examine alcoholism is very interesting. Unfortunately, it is problematic in that the terms used are confusing, the derived metrics are poorly justified, and independent verification was generally not done. To better clarify these criticisms:

We thank the reviewer for supporting our work and the insightful and constructive comments.

Confusing terms:

1. Prior studies seem to refer to “supply” as CMRglc and “demand” as IFCD. I would recommend using the original terms. Regarding supply, this study itself acknowledges aerobic glycolysis which (when the “glycolytic index” measure is examined) shows that oxygen metabolism and glucose metabolism can be very different. Regarding demand, some of the authors’ prior work has shown that IFCD does not cover every form of fluctuation measurable with resting state fMRI in its relationship to CMRglc.

We agree with the reviewer that CMRglc and IFCD do not fully capture all supply and demand processes in the brain. In the revised manuscript, when specifically referring to synchronous BOLD fluctuations and cerebral glucose metabolism, we use “IFCD” and “CMRglc”, respectively. We use “demand” and “supply” terms only for referring to generic demand and supply processes in the brain. In addition, we have explicitly highlighted that IFCD does not capture all forms of measurable resting state fluctuations. For example:

“We used this characterization of the brain’s CMRglc-IFCD dynamics (indexing important components of neuronal activity demand and metabolic supply) to classify the brain into major segments based on rPWR and rCST.”

“Here we used CMRglc (indexing brain’s main energy supply) and IFCD (indexing aspects of neuroglial activity) and defined novel measures of rPWR (extent of concurrently high CMRglc and IFCD) and rCST (the extent to which CMRglc leads IFCD) to study the variations in the coupling between CMRglc-IFCD across brain regions.”

“Specifically, in a two-dimensional map of (mean-variance normalized) CMRglc-IFCD (Fig. 2e) we performed...”

“While some metrics focus on regional synchrony in slow (< 0.1 Hz) BOLD fluctuations as a marker of neuronal activity (e.g., IFCD¹ or regional homogeneity measures²), others focus on the amplitude of slow fluctuations (e.g., ALFF or fALFF) during resting state^{3,4}. Synchrony-based measures only capture aspects of resting-state activity, however, synchrony-based measures and amplitude-based measures have shown good correspondence at rest⁵ and between different subject groups⁶.”

2. “Power” already has a commonly used meaning in time series, and as this study uses fMRI time series to calculate IFCD, its use here is confusing. “Cost” has clearer meaning here (than supply, demand, or power) but ignores that glucose may be being used for purposes other than IFCD, causing the change in “cost.”

Following the reviewer’s advice, we renamed these metrics to highlight that they are *relative* and now use distinct acronyms (i.e., rPWR and rCST) throughout the text.

“To characterize this coupling, here we defined novel measures of relative power (rPWR, extent of concurrently high activity and energy utilization) and relative cost (rCST, extent that energy utilization exceeds activity).”

We distinguished rPWR from its alternative uses in the literature. We also discuss that these metrics are relevant here for when CMRglu and IFCD measures are used to represent supply/demand in the brain (please also see response to comment 7 for validity of rPWR/rCST using alternative metrics/modalities). rPWR was inspired by the definition of power (P) in electrical circuits (but rPWR does not represent P), where instantaneous $P = VI$, where “V” represents electrical potential (analogous to activity demand) and “I” represents electrical current (analogous to metabolic supply).

“It is worth noting that rPWR and rCST values are relative measures (to the rest of the brain) with the assumption that CMRglc indexes glucose energy supply (among other substrates) and that synchronous regional fluctuations in the BOLD signal are proportional to local activity, where each capture a fraction of energy expenditure and neuroglial activity, respectively.”

“rPWR should not be confused with other uses of term “power” in the literature such as power in electrical circuits (though they bare some similarity), statistical power, and power in time series.”

We agree with the reviewer that the definition of rCST does not preclude uses of glucose for purposes other than IFCD. This is clarified in the discussion.

“Higher rCST could be attributed to use of less efficient (but fast) glucose metabolic pathways (e.g., aerobic glycolysis)^{7,8}, or even a higher glia to neuron ratio in the neo cortex⁹. In fact, our estimate of regional rCST (Fig. 2h) had a good correspondence with previously reported distribution of aerobic glycolysis (accounting for about 10% of glucose metabolized by the adult brain)⁷, which highlights dorsal-medial frontal, precuneus, posterior cingulate and lateral frontal regions (see Supplementary Table 2).”

“Lower rCST in regions such as cerebellum (Fig. 2h and Supplementary Table 2) could be attributed to higher regional proportion of oxidative phosphorylation to aerobic glycolysis⁷ or use of energy sources other than glucose such as ketone bodies.”

Poor justification of derived metrics:

3. *Prior studies by some of the authors and others well-characterized both similarities and differences between CMRglc and IFCD (and other fMRI measures). What is gained by the transformation from supply/demand to cost/power? Most tests are only done on supply/demand but not cost/power. Do cost/power provide fundamentally different information? They are further derived and thus harder to conceptualize measures, for this reason the authors need to justify what they provide that the less derived measures do not.*

We thank the reviewer for the important suggestion. We have highlighted in the text the utility of the new measures of rCST and rPWR.

“Notably, variations in how energy is supplied (and metabolized) in different brain regions (spatially)⁸ and under different stimulation and physiological conditions (temporally)¹⁰ are of high relevance in our understanding of brain physiology¹¹, development⁸, cognitive abilities¹², and neuropsychiatric disorders^{13, 14}. However, less attention has been paid to regional variations in metabolic supply (e.g., glucose metabolism) while accounting for the underlying activity. The correspondence between neuroglial activity and glucose metabolism is important for characterizing functional specialization of brain networks, yet this correspondence has not been well quantified in the literature.”

“The associations between activity demand and metabolic supply in the brain are important for studying brain function^{8, 11, 12} and diseases^{13, 14}, yet these associations have not been well quantified in the literature. In principle, energy demand and supply in the brain are matched (with certain exceptions such as heat¹⁵ or lactate¹⁶ production). However, our measures of activity demand (i.e., IFCD) and metabolic supply (i.e., CMRglc) only capture specific aspects of demand and supply processes in the brain, making their mismatch not only possible but also informative. Conditions such as exposure to alcohol¹⁷ or sleep¹⁰ are known to impact brain glucose metabolism due to the use of alternative source of energy (e.g., acetate) or changes in active glucose metabolic pathways (aerobic glycolysis versus oxidative metabolism). However, without accounting for underlying brain activity, changes in glucose metabolism are hard to interpret, especially when measures of activity and energy metabolism are studied in isolation. To quantify the relative changes in the association between two modalities, here we proposed two novel metrics of rPWR and rCST that are unit-free and are generalizable to any measures of brain activity and energy supply. These metrics quantify how well two modalities that measure aspects of activity demand (such as IFCD) and aspects metabolic supply (such as CMRglc) are concurrently high or low (i.e., rPWR) and to what extent one exceeds the other (i.e., rCST) relative to the rest of the brain. From another perspective, rPWR could be thought of as an index of *loudness* with respect to any of the two modalities, while rCST is an index of *mismatch* of the two modalities in each data point. Since rPWR and rCST are relative, they are not sensitive to global changes in any of the two modalities (e.g., levels of glucose metabolism as in acute or chronic alcohol exposure, Fig. 5), and could

be useful to map and track the associations between specific markers of activity demand and metabolic supply under different physiological (e.g., sleep), pharmacological (e.g., alcohol intoxication), or disease (e.g., AD) states. Analysis of individual differences showed a clear segregation of different brain networks based on rPWR and rCST (Fig. 3f) relative to IFCD and CMRglc (Fig. 3c, also see Supplementary Table 8). We found that rPWR and rCST are robust and generalizable to other measures of activity demand and metabolic supply (e.g., fALFF and CBF, Supplementary Figs. 4, 5) and do not appear to amplify the effects of measurement noise on IFCD and CMRglc (Supplementary Tables 11, 12). Analysis of whole-brain distribution of rPWR and rCST characterized alterations in the IFCD-CMRglc coupling due to acute alcohol exposure and showed that prolonged alcohol use may shift the brain toward less efficient energetic states. More importantly, we found that rPWR and rCST (relative to IFCD and CMRglc) were significantly and distinctly related to behavioral effects of acute and chronic alcohol, further supporting their utility for capturing meaningful (and possibly unique) aspects of brain performance. Thus, we propose that rPWR and rCST as new multimodal metrics to study the energetic economy of brain networks¹⁸ throughout the lifespan and to monitor the effects of drugs and diseases on the human brain.”

4. Why is the transformation sinusoidal instead of a projection or coordinate transformation? This seems to create a gap in the cloud distribution (Figure 1D/F). Is this desirable? Why? Why does it then look like a rotation in 2D space (Figure 2A/B), which hypothetically shouldn't matter for clustering?

We have now clarified this confusion in the text. The rPWR/rCST transformation is indeed a 45-degree rotation of axes in the 2D space of mean-variance normalized CMRglc-IFCD axes. This has also been described in Fig. 4b, c (former Fig. 2a, b) where the segmentations are depicted along the rCST-rPWR and CMRglu-IFCD axes. The formula presented in the Methods show this effect in a polar coordinate system which mirrors our implementation, but now we also include the cartesian equivalent of this transformation. It is important to note that the plots in Fig. 2d/f (former Fig. 1d/f) only highlight voxels which contribute the most to rPWR variability (now Fig. 2d) and to rCST variability (now Fig. 2f).

“Highlighting” was only performed for visual demonstration by multiplying the radius of each voxel (in polar coordinate system) by its corresponding absolute rPWR (Fig. 2d) or absolute rCST (Fig. 2f). This is now clarified in the figure caption. It is noteworthy that without mean-variance normalization of CMRglc-IFCD (which is done in the process of rPWR/rCST transformation), the segmentation will be primarily driven by regional differences in CMRglc, considering that CMRglc has higher range than IFCD.

“rPWR and rCST. Voxelwise relative power (rPWR) and relative cost (rCST) were computed by a $\pi/4$ rad (45°) counterclockwise rotation of (mean and variance normalized) $\log(\text{IFCD})$ and CMRglc axes, respectively. Specifically, in a two-dimensional polar coordinate system of standardized (whole-brain mean = 0, variance = 1) $\log(\text{IFCD})$, $z(\log(\text{IFCD}))$, plotted against standardized (whole-brain mean = 0, variance = 1) CMRglc, $z(\text{CMRglc})$, we define:

$$rPWR = R \times \cos\left(\theta - \frac{\pi}{4}\right),$$

and

$$rCST = R \times \sin\left(\theta - \frac{\pi}{4}\right),$$

where R and ϑ are radius and angle (in radians) of each brain voxel in a polar coordinate system of $z(\log(\text{IFCD}))$ and $z(\text{CMRglc})$, respectively. Alternatively, in a cartesian coordinate system,

$$\begin{bmatrix} rPWR \\ rCST \end{bmatrix} = \begin{bmatrix} \cos \frac{\pi}{4} & \sin \frac{\pi}{4} \\ -\sin \frac{\pi}{4} & \cos \frac{\pi}{4} \end{bmatrix} \begin{bmatrix} z(\log(\text{IFCD})) \\ z(\text{CMRglc}) \end{bmatrix}.$$

“Fig. 2... (d–f) rPWR and rCST were calculated by a $\pi/4$ (45°) rotation along (mean-variance normalized) $\log(\text{IFCD})$ and CMRglc axes. **(e)** A hypothetical presentation of (mean-variance normalized) activity demand versus metabolic supply with each circle representing one brain voxel. Yellow-colored voxels correspond to higher rCST, blue to lower rCST, red to higher rPWR, and green to lower rPWR. For a representative voxel v_i (dark gray circle), $rPWR_i$ and $rCST_i$ are shown on the plot. For v_i , rPWR is negative and rCST is positive. For visual demonstration purposes, voxels contributing the most to rPWR variability **(d)** and rCST variability **(f)** are highlighted. For this purpose, *highlighting* was performed by multiplying the radius of each voxel (in polar coordinate system) by its corresponding absolute rPWR **(d)** or absolute rCST **(f)**. Group-average rPWR **(g)** and rCST **(h)** maps.”

5. The major focus of the study is power/cost, so why is the stimulation data for which PET appear to be unavailable used? If power and cost provide information supply and demand do not, then this isn't comparable as power and cost cannot be calculated without supply data.

The visual stimulation data were only used to show the sensitivity of IFCD relative to fALFF to activity at 0.05 Hz (comparable to major resting-state frequencies). We have now clarified the purpose of the visual stimulation experiment and placed the experiment at the beginning of Results.

“In cohort-vs (vs: visual stimulation; $n = 7$ healthy participants) with fMRI data only, we tested the sensitivity of IFCD to brain activity, by measuring the effects of visual stimulation ($f = 0.05$ Hz) on IFCD relative to fractional amplitude of low frequency fluctuations (fALFF) in the visual cortex.”

“Functional significance of IFCD. To corroborate the validity of IFCD as a measure of neuronal activity, we assessed the effects of a visual stimulation at $f = 0.05$ Hz on IFCD relative to fALFF in cohort-vs ($n = 7$).”

Independent verification not done:

6. This is a multimodal study and, while prior work has examined the effect of PET and fMRI being recorded through very different methods, this study calculates new measures from them. When new measures are derived, small portions of the variance in source data could be amplified. Thus it is critical to validate that the new measures are not amplifying noise. One area in particular is the visual cortex which appears strongly in this study with cost/power but can often have good signal in PET but poor signal in fMRI. At a minimum, the cost/power calculation and SPM and k-means clustering (same number of clusters) should be done on SNR maps for PET and fMRI to see if the SNR variation between PET and fMRI causes some of the clusters.

This is an important concern. We calculated the temporal SNR (tSNR) maps for fMRI and PET data (Supplementary Tables 10, 11). For example, for both fMRI and PET we found that tSNR was in the mid to high range in different segments of visual cortex. As the reviewer advised, we repeated the clustering for rPWR and rCST measures based on fMRI and PET tSNR maps. We also characterized how individual differences in rPWR and rCST (Supplementary Table 8) are related to tSNR in different regions (Supplementary Tables 11, 12). Overall, rPWR and rCST did not appear to amplify the effects of tSNR. These analyses are now updated in the Results and Methods:

“For fMRI, tSNR (see Methods) was overall negatively associated with IFCD ($p = 0.005$) and positively with CMRglc ($p = 0.0004$) across regions (Supplementary Table 11), but not with rPWR nor with rCST. For PET-FDG, tSNR (see Methods) was negatively associated with IFCD ($p = 0.02$) and positively with CMRglc ($p < 0.0001$) (Supplementary Table 12). PET-FDG tSNR was also significantly associated with rPWR ($p = 0.02$) and rCST ($p = 0.03$), but both to a weaker extent than the associations between PET-FDG tSNR and CMRglc ($p < 0.0001$, Supplementary Table 12). While temporal pole and entorhinal regions both has the lowest tSNR in PET-FDG and fMRI, their rPWR and rCST were not associated with tSNRs (Supplementary Tables 11, 12). In fact, rPWR and rCST in none of the cortical regions were associated with tSNRs after correction for multiple comparisons. We also used PET-FDG tSNR and fMRI tSNR maps to compute SNR-based rPWR and rCST maps and performed cortical segmentation with k -means clustering ($k = 4$, Supplementary Fig. 6) and found that the clusters differed from those based on CMRglc-IFCD.”

“Effects of Brain morphometry and temporal signal-to-noise ratio (tSNR)...For fMRI, voxelwise tSNR was computed using the mean to standard deviation ratio of the raw fMRI time series (after motion correction and spatial normalization). For PET-FDG, voxelwise tSNR was computed using the mean to standard deviation ratio of the dynamic FDG time series (after motion correction and spatial normalization, acquired between 40–55 min). For each subject, each tSNR measure was averaged within each of the 34 bilateral ROIs.”

“However, we cannot rule out that inherent limitations in PET and MRI imaging (e.g., spatial heterogeneity in tSNR) could to some extent affect regional differences in rCST and rPWR. We observed that rPWR and rCST do not appear to amplify effects of tSNR in fMRI and PET-FDG (Supplementary Tables 11, 12 and Supplementary Fig. 6). Visual networks showed significant differences in rCST and rPWR (Fig. 3d, e; Fig. 4a), yet they had mid-to-high range tSNR in both modalities (Supplementary Tables 11, 12), suggesting that differences in rPWR and rCST in visual regions are not primarily related to measurement noise.”

7. Cost and power would be more justifiable (versus the existing measures of CMRglc and IFCD) if they were independently verified where existing measures were not. E.g. the authors could use EEG to measure demand and MRS to measure supply in only two widely separated brain regions with high cost/power deviation. There are many other possible ways to test this, but demonstration that new metrics have validity outside of combined PET/fMRI would go a long way to validating them.

As suggested by the reviewer, we performed an independent verification of rPWR and rCST. Accordingly, we used a relative measure of cerebral blood flow (CBF) (with perfusion weighted imaging, PWI) as an alternative modality to PET-FDG to index metabolic supply. We also used fLAFf as an alternative metric of activity which in contrast to IFCD (sensitive to synchrony between voxels) is sensitive to amplitude

changes in voxels. rPWR and rCST estimated based on CBF-fALFF were in excellent agreement with rPWR and rCST estimated based on CMRglc-IFCD (Supplementary Figs. 3–5):

“Generalizability of rPWR and rCST. We tested whether rPWR and rCST are generalizable to alternative measures of neuronal activity (i.e., fALFF) and metabolic supply (i.e., CBF) in cohort-1. Based on prior observations, we used cerebral blood flow (CBF) as an alternative proxy of cerebral metabolic supply^{19, 20}. Consistent with prior reports (Supplementary Fig. 3a)¹⁹, CMRglc and CBF (PWI) (see Methods) showed a good correspondence across brain networks in our study (Supplementary Fig. 3b). IFCD and fALFF were also highly correlated across networks (Supplementary Fig. 3c) but they were not significantly correlated with CMRglc and CBF (PWI) (Supplementary Fig. 1a, Supplementary Fig. 3d–f). There was excellent agreement between rPWR of different networks when estimated using CMRglc-IFCD and when estimated using CBF-fALFF (ICC(3, 1) = 0.8, Supplementary Fig. 4a). While only using CBF (PWI) instead of CMRglc also showed strong agreement in rPWR estimates (ICC(3, 1) = 0.85, Supplementary Fig. 4c). The IFCD threshold had minimal impact on rPWR estimates (ICC(3, 1) = 0.99, Supplementary Fig. 4d). Similarly, there was excellent agreement between rCST of different networks when estimated using CMRglc-IFCD and when estimated using CBF-fALFF (ICC(3, 1) = 0.87, Supplementary Fig. 5a). Changing IFCD to fALFF had minimal impact on rCST estimates (ICC(3, 1) = 0.99, Supplementary Fig. 5b), so did using CBF (PWI) instead of CMRglc (ICC(3, 1) = 0.85, Supplementary Fig. 5c). The IFCD threshold had minimal impact on rCST estimates, as well (ICC(3, 1) = 0.99, Supplementary Fig. 5d).”

“We found that rPWR and rCST are robust and generalizable to other measures of activity demand and metabolic supply (e.g., fALFF and CBF, Supplementary Figs. 4, 5) and do not appear to amplify the effects of measurement noise on IFCD and CMRglc (Supplementary Tables 11, 12).”

8. Voxel-based morphometry (VBM) is commonly used in alcohol use and alcoholism. The relative gray matter/white matter levels are thus of suspicion when comparing PET and fMRI modalities. (The authors limit their analysis to gray matter, but VBM applies within gray matter alone.) Comparing the cost/power significance versus VBM significance would also help validate these measures as something novel. (Though if VBM differences and cost/power differences are too similar it is unknown whether activity is different due to different anatomy, or if anatomy creates an artifact which alters activity measurement.)

Following the reviewers' comment we now report effects of brain morphometry on rCST and rPWR. Unfortunately, the anatomical data in cohort-2 (or BNL cohort, for which we studied the effects of alcohol) was of poor quality for reliable VBM analysis. Considering that surface based morphometry (SBM) versus voxel based morphometry (VBM) provide comparable results²¹ and to keep the processing pipelines consistent between analyses, we used measures of cortical thickness and cortical distance (as also suggested by another reviewer) and report effects of brain morphometry on rCST and rPWR.

“We studied how measures of brain morphometry (i.e., cortical thickness and cortical distance), and fMRI tSNR and PET-FDG tSNR were related to IFCD, CMRglc, rPWR, and rCST. Specifically, these measures were related across subjects in 34 bilateral cortical ROIs (see Methods) in cohort-1. Across-subject mean and standard deviation in IFCD, CMRglc, rCST, and rPWR in these regions are summarized in Supplementary Table 8. Supplementary Tables 9–12 show regional associations with tSNRs while also reporting level of significance for the correlations across regions. Across regions, there were a positive

trend of correlations between cortical thickness and IFCD ($p < 0.0001$) and between cortical thickness and CMRglc ($p < 0.0001$), but not with rPWR nor with rCST. rPWR in superior frontal region was associated with cortical thickness ($p < 0.05$, Bonferroni). Cortical distance (average geometrical distance between an ROI to other ROIs, see Methods) showed a significant trend of negative correlation across regions with CMRglc (Supplementary Table 10). Cortical distance was associated with rCST in entorhinal, lateral orbitofrontal, and inferior temporal cortices and with rPWR in pars triangularis ($p < 0.05$, Bonferroni).”

“We found that average cortical thickness and cortical distance in several regions were positively associated with rCST and rPWR ($p < 0.05$, Bonferroni, Supplementary Tables 9, 10). It could be expected that higher cortical distance (or thickness) is consistent with greater activity and energetic needs, particularly for regions with significant remote connectivity. Interestingly, cortical distance and rCST were associated in entorhinal, lateral orbitofrontal, and inferior temporal cortices, which are among regions implicated in Alzheimer’s diseases (AD)^{22, 23}. Nevertheless, the relevance of these morphological findings remains to be determined.”

The noted problems are addressable but require a substantial rewrite with substantial new data analysis.

Reviewer #2 (Remarks to the Author):

The authors acquired BOLD-fMRI and FDG-PET data from healthy subjects and devised a novel, integrated measure of power/cost to identify regions with high/low fMRI activity at high levels of energy utilization. They found distinct power-cost distributions across the cortex of healthy subjects with characteristic changes in another cohort of alcohol drinkers.

While I found it highly interesting to gain novel insights into human brain organization from multimodal imaging, I have major concerns with the methodology of defining this novel measure of power/cost and was less convinced by the extended view about brain organization derived from the novel approach.

We thank the reviewer for the constructive and important criticisms.

The measure of IFCD is a tightly defined parameter to capture neuronal activity from the BOLD-fMRI signal. It captures the synchrony of BOLD-signal fluctuations between a specific voxel and voxels in close neighborhood at a particular threshold C which is then log-transformed. In order to relate fMRI-derived neuronal activity to absolute glucose metabolism, I would suggest to offer a broader perspective on BOLD-measures. How do parameters used in this study such as thresholding, log-transformation, pearson-correlation instead of signaling amplitude impact on the relation of IFCD to CMRglc...

We have now provided a broader perspective on IFCD in the Discussion (and partly in the Introduction). Following other reviewer's suggestion, we renamed power/cost to rPWR/rCST to minimize any confusion with other metrics. We studied the effects of IFCD threshold and amplitude-based measures of resting-state activity on the relationship between CMRglc and resting-state activity (Supplementary Figs. 3–5). We also discuss the effect of using log-transformation for IFCD (Supplementary Fig. 7):

“In cohort-vs (vs: visual stimulation; $n = 7$ healthy participants) with fMRI data only, we tested the sensitivity of IFCD to brain activity, by measuring the effects of visual stimulation ($f = 0.05$ Hz) on IFCD relative to fractional amplitude of low frequency fluctuations (fALFF) in the visual cortex.”

“We tested whether rPWR and rCST are generalizable to alternative measures of neuronal activity (i.e., fALFF) and metabolic supply (i.e., CBF) in cohort-1. Based on prior observations, we used cerebral blood flow (CBF) as an alternative proxy of cerebral metabolic supply^{19,20}. Consistent with prior reports (Supplementary Fig. 3a)¹⁹, CMRglc and CBF (PWI) (see Methods) showed a good correspondence across brain networks in our study (Supplementary Fig. 3b). IFCD and fALFF were also highly correlated across networks (Supplementary Fig. 3c) but they were not significantly correlated with CMRglc and CBF (PWI) (Supplementary Fig. 1a, Supplementary Fig. 3d–f). There was excellent agreement between rPWR of different networks when estimated using CMRglc-IFCD and when estimated using CBF-fALFF (ICC(3, 1) = 0.8, Supplementary Fig. 4a). While only using CBF (PWI) instead of CMRglc also showed strong agreement in rPWR estimates (ICC(3, 1) = 0.85, Supplementary Fig. 4c). The IFCD threshold had minimal impact on rPWR estimates (ICC(3, 1) = 0.99, Supplementary Fig. 4d). Similarly, there was excellent agreement between rCST of different networks when estimated using CMRglc-IFCD and when estimated using CBF-fALFF (ICC(3, 1) = 0.87, Supplementary Fig. 5a). Changing IFCD to fALFF had minimal impact on rCST estimates (ICC(3, 1) = 0.99, Supplementary Fig. 5b), so did using CBF (PWI) instead of CMRglc (ICC(3, 1) = 0.85, Supplementary Fig. 5c). The IFCD threshold had minimal impact on rCST estimates, as well (ICC(3, 1) = 0.99, Supplementary Fig. 5d).”

“PET-FDG is a reliable method for measuring CMRglc in the human brain. In comparison, there are a range of metrics available to assess voxel-level functional activity during resting-state fMRI. While some metrics focus on regional synchrony in slow (< 0.1 Hz) BOLD fluctuations as a marker of neuronal activity (e.g., IFCD¹ or regional homogeneity measures²), others focus on the amplitude of slow fluctuations (e.g., ALFF or fALFF) during resting state^{3,4}. While synchrony-based and amplitude-based measures appear capture different aspects of resting-state activity, they have shown good correspondence at rest⁵ and between different subject groups⁶.”

“Using CBF (PWI) and fALFF as alternative indices of metabolic supply and neuronal activity resulted in rPWR and rCST that had strong agreement with those obtained with CMRglc and IFCD (Supplementary Figs. 4, 5) further supporting the generalizability of these metrics.”

“Because degree-related measures (such as IFCD) follow an exponential distribution, we used $\log(\text{IFCD})$ in all analyses with a semi-normal distribution (Supplementary Fig. 7) to characterize brain activity and its associations with brain glucose metabolism (indexed by CMRglc).”

and do other parameters such as structural distance or cortical thickness possibly relate to the regional distribution of power/cost?

We now added a section on the relationship between brain morphometry and rPWR and rCST:

“We studied how measures of brain morphometry (i.e., cortical thickness and cortical distance), and fMRI tSNR and PET-FDG tSNR were related to IFCD, CMRglc, rPWR, and rCST. Specifically, these measures were related across subjects in 34 bilateral cortical ROIs (see Methods) in cohort-1. Across-subject mean and standard deviation in IFCD, CMRglc, rCST, and rPWR in these regions are summarized in Supplementary Table 8. Supplementary Tables 9–12 show regional associations with tSNRs while also reporting level of significance for the correlations across regions. Across regions, there were a positive trend of correlations between cortical thickness and IFCD ($p < 0.0001$) and between cortical thickness and CMRglc ($p < 0.0001$), but not with rPWR nor with rCST. rPWR in superior frontal region was associated with cortical thickness ($p < 0.05$, Bonferroni). Cortical distance (average geometrical distance between an ROI to other ROIs, see Methods) showed a significant trend of negative correlation across regions with CMRglc (Supplementary Table 10). Cortical distance was associated with rCST in entorhinal, lateral orbitofrontal, and inferior temporal cortices and with rPWR in pars triangularis ($p < 0.05$, Bonferroni).”

“We found that average cortical thickness and cortical distance in several regions were positively associated with rCST and rPWR ($p < 0.05$, Bonferroni, Supplementary Tables 9, 10). It could be expected that higher cortical distance (or thickness) is consistent with greater activity and energetic needs, particularly for regions with significant remote connectivity. Interestingly, cortical distance and rCST were associated in entorhinal, lateral orbitofrontal, and inferior temporal cortices, which are among regions implicated in Alzheimer’s diseases (AD)^{22, 23}. Nevertheless, the relevance of these morphological findings remains to be determined.”

It is important to note that the novel power/cost measure was only derived from N=17 healthy subjects of a total of N=75.

We clarified the number of subjects for each experiment and used a new labelling to identify different cohorts in the study. rPWR and rCST were derived with high resolution PET-FDG and fMRI data in cohort-1 ($n = 28$). We also referred to the relevant cohort in each figure. We apologize for any confusion.

“For this purpose, we performed a series of experiments and analyses in three independent cohorts. In cohort-vs (vs: visual stimulation; $n = 7$ healthy participants) with fMRI data only, we tested the sensitivity of IFCD to brain activity...”

“In cohort-1 ($n = 28$ healthy participants) with high-resolution PET-FDG and fMRI, voxelwise measures of relative power (rPWR; indicating high metabolic needs and observed activity relative to the rest of the brain) and relative cost (rCST; the extent to which regional metabolic needs exceeds the observed activity) were computed.”

“Finally, in cohort-2 ($n = 40$) with PET-FDG and fMRI, we tested the sensitivity of rPWR and rCST to acute and chronic alcohol exposure which affect brain glucose metabolism^{17, 24, 25} and neuronal activity^{26, 27}, in light drinkers (LD, $n = 24$) and a heavy drinkers (HD, $n = 16$) (see Methods).”

From those 17 subjects, was the PET data always related to the fMRI from the same day or possibly from the second MRI scan?

The PET-FDG data were related to IFCD (or fALFF) data (in cohort-1, $n = 28$) from the average of the two fMRI sessions, which is now clarified in the text.

“For cohort-1, the average $\log(\text{IFCD})$ from the two fMRI sessions were used in the analyses to improve signal to noise ratio of activity measures. In cohort-1, we tested the sensitivity of rPWR and rCST to IFCD threshold by using $r = 0.4$ as an alternative IFCD threshold (compared to $r = 0.6$).”

“For cohort-1, the average fALFF from the two fMRI sessions were used in the analyses to improve signal to noise ratio of activity measures.”

The authors report a strong linear correlation between $\log(\text{IFCD})$ and CMRglc (Suppl Fig 1) which I found not interpretable with respect to the raw data presented in Fig 1B of the main manuscript. The heat plot in Fig 1B rather suggests a very confined and heavy aggregation of voxels towards the lower left end of the graph with almost no variance along the $\log(\text{IFCD})$ axis. With respect to the slope in Suppl Fig 1 this linear correlation is hardly justified when plotting onto the heatmap data.

We removed this confusing figure and replaced it by Supplementary Fig 1a which summarizes IFCD and CMRglc averages and their association across 10 brain networks.

“As in CMRglc and IFCD (Supplementary Fig. 1a), and rPWR and rCST were not correlated across networks (Supplementary Fig. 1b), supporting the heterogeneity of these measures across brain networks.”

Moreover, I did not understand how the regular/circular distribution of power/cost from the model (Fig 1E) would fit onto the heavy tailed voxel distribution in Fig 1B. It seems that >90% of all voxels would fall into the low power-quadrant making it hard to interpret whole cortex data.

We have clarified in the figure caption (and the text) that the axes in the hypothetical model are mean-variance normalized (as they are prior to rPWR and rCST transformations) but the data shown in Fig. 2b (previously, Fig. 1b) are in the original IFCD and CMRglc space (thus the centers of axes in Fig. 3b do not represent means). The ratio of voxels falling into each quadrant (relative to mean IFCD and CMRglc) is now estimated and displayed in Fig. 4c which suggests a relatively balanced distribution. For example, 36% of voxels fall into the low-power quadrant.

“Fig. 2.... (d–f) rPWR and rCST were calculated by a $\pi/4$ (45°) rotation along (mean-variance normalized) $\log(\text{IFCD})$ and CMRglc axes. (e) A hypothetical presentation of (mean-variance normalized) activity demand versus metabolic supply (not to be confused with part (b) shown without mean-variance normalization) with each circle representing one brain voxel. Yellow-colored voxels correspond to higher rCST, blue to lower rCST, red to higher rPWR, and green to lower rPWR. For a representative voxel v_i (dark gray circle), rPWR_i and rCST_i are shown on the plot. For v_i , rPWR is negative and rCST is positive. For visual demonstration purposes, voxels contributing the most to rPWR variability (d) and rCST variability (f) are highlighted. For this purpose, highlighting was performed by multiplying the radius of

each voxel (in polar coordinate system) by its corresponding absolute rPWR (d) or absolute rCST (f). Group-average rPWR (g) and rCST (h) maps.”

“Fig. 4....(c) Same clusters projected back into the original CMRglc-IFCD space. The thick gray lines mark the average of CMRglc and average of log(IFCD) with the number in each quadrant showing percentage of voxels falling within that quadrant. These quadrants are defined in Fig. 2e.”

Together, this critique is also reflected by the fact that the novel measure of power/cost (Fig 1 G/H) reveals an almost identical spatial distribution as CMRglc (Fig 1C) alone which indeed has strongest variance in Fig 1B. Several voxels seem to have highest relative levels of both power and cost (lateral prefrontal regions). Doesn't that contradict each other?

Since rPWR and rCST were calculated based on mean-variance normalized IFCD and CMRglc, both measures contributed to variance in rPWR and rCST. We studied this further and reported results in the section on **rPWR and rCST of resting state networks and individual differences** and in Fig. 3 which compares IFCD, CMRglc, rPWR, and rCST across 10 brain networks (please also see Supplementary Table 8 for individual differences in regions). While mean IFCD was more variable between the 3 visual and default mode networks, mean CMRglc was more variable across other networks. Across the 10 network, rPWR and rCST were related to *both* modalities: IFCD-rPWR: $r(8) = 0.82$, IFCD-rCST: $r(8) = -0.7$; CMRglc-rPWR: $r(8) = 0.74$, CMRglc-rCST: $r(8) = 0.53$. Due to limited dynamic range in colors in Fig. 2g, h (former Fig. 1g, h), Fig. 3 could help to better show regional differences in rPWR and rCST (we clarified this in Fig. 2). For example, while both left and right frontoparietal networks had the highest rCST, they had intermediate rPWR.

“Left and right frontoparietal networks had the highest rCST, whereas medial visual followed by the cerebellum and occipital pole networks had the lowest rCST.”

“Since rPWR and rCST were calculated from mean-variance normalized IFCD and CMRglc, both measures contributed to variance in rPWR and rCST (see Methods).”

“Alternatively, in a cartesian coordinate system,

$$\begin{bmatrix} rPWR \\ rCST \end{bmatrix} = \begin{bmatrix} \cos \frac{\pi}{4} & \sin \frac{\pi}{4} \\ -\sin \frac{\pi}{4} & \cos \frac{\pi}{4} \end{bmatrix} \begin{bmatrix} z(\log(IFCD)) \\ z(CMRglc) \end{bmatrix}.”$$

“Fig. 2....Group-average rPWR (g) and rCST (h) maps but see Fig. 3 (and Supplementary Table 8) for demonstration of regional differences.”

“We studied how measures of brain morphometry (i.e., cortical thickness and cortical distance), and fMRI tSNR and PET-FDG tSNR were related to IFCD, CMRglc, rPWR, and rCST. Specifically, these measures were related across subjects in 34 bilateral cortical ROIs (see Methods) in cohort-1. Across-subject mean and standard deviation in IFCD, CMRglc, rCST, and rPWR in these regions are summarized in Supplementary Table 8.”

Overall, I found the research question and concept of integrating both imaging modalities very interesting. However, I don't see the major claims supported by the data and would suggest a more detailed derivation of the complex measures power/cost from the fMRI data.

As the reviewer suggested, we performed multiple analysis to test the generalizability (i.e., estimating rPWR and rCST with alternative metrics) and robustness (i.e., studying the effects of brain morphometry and tSNR) of rPWR and rCST measures. For equal contribution IFCD and CMRglc, we also clarified that both modalities are mean-variance normalized prior to rPWR and rCST transformations.

Reviewer #3 (Remarks to the Author):

In this manuscript, the authors have defined novels measures for characterizing coupling between glucose metabolism (measures with PET imaging) and resting state connectivity (measured with fMRI). They came up with two measures; power (extent of high activity and energy utilization) and cost (the extent to which energy utilization exceeds activity), and showed that different regions of the brain has different ratios of these measures. They also showed that these measures may be altered in heavy alcohol drinkers, and with acute alcohol administration.

We thank the reviewer for very helpful and insightful suggestions.

While this is an interesting concept, questions remain about how robust and reliable these measures are:

Following the reviewer suggestion (as well as others) we performed additional analyses and computed rPWR and rCST (formerly known as "power" and "cost") with alternative measures of brain activity and metabolic supply:

"Generalizability of rPWR and rCST. We tested whether rPWR and rCST are generalizable to alternative measures of neuronal activity (i.e., fALFF) and metabolic supply (i.e., CBF) in cohort-1. Based on prior observations, we used cerebral blood flow (CBF) as an alternative proxy of cerebral metabolic supply¹⁹,²⁰. Consistent with prior reports (Supplementary Fig. 3a)¹⁹, CMRglc and CBF (PWI) (see Methods) showed a good correspondence across brain networks in our study (Supplementary Fig. 3b). IFCD and fALFF were also highly correlated across networks (Supplementary Fig. 3c) but they were not significantly correlated with CMRglc and CBF (PWI) (Supplementary Fig. 1a, Supplementary Fig. 3d–f). There was excellent agreement between rPWR of different networks when estimated using CMRglc-IFCD and when estimated using CBF-fALFF (ICC(3, 1) = 0.8, Supplementary Fig. 4a). While only using CBF (PWI) instead of CMRglc also showed strong agreement in rPWR estimates (ICC(3, 1) = 0.85, Supplementary Fig. 4c). The IFCD threshold had minimal impact on rPWR estimates (ICC(3, 1) = 0.99, Supplementary Fig. 4d). Similarly, there was excellent agreement between rCST of different networks when estimated using CMRglc-IFCD and when estimated using CBF-fALFF (ICC(3, 1) = 0.87, Supplementary Fig. 5a). Changing IFCD to fALFF had minimal impact on rCST estimates (ICC(3, 1) = 0.99, Supplementary Fig. 5b), so did using CBF (PWI) instead of CMRglc (ICC(3, 1) = 0.85, Supplementary Fig. 5c). The IFCD threshold had minimal impact on rCST estimates, as well (ICC(3, 1) = 0.99, Supplementary Fig. 5d)."

“We found that rPWR and rCST are robust and generalizable to other measures of activity demand and metabolic supply (e.g., fALFF and CBF, Supplementary Figs. 4, 5) and do not appear to amplify the effects of measurement noise on IFCD and CMRglc (Supplementary Tables 11, 12).”

whether they are highly variable between individuals,

We also updated a section and clarified in Fig. 3 individual differences in rPWR and rCST and also added Supplementary Table 8 which list variability in rPWR and rCST across individuals. We also assessed how individual differences in rPWR and rCST are related to brain morphometry (Supplementary Tables 9, 10).

“**rPWR and rCST of resting state networks and individual differences.** Fig. 3a, b show individual differences in CMRglc and IFCD within 10 predefined resting-state network maps²⁸. Plotting these values against each other (Fig. 3c, Supplementary Fig. 1a) shows low segregation of these networks particularly when considering individual differences. Similarly, we compared individual differences in rPWR and rCST of these 10 networks (Fig. 3d, e).”

“Fig. 3f shows subject-level averages of rCST and rPWR when plotted against each other, which in contrast to Fig. 3c, highlighted consistency of rCST and rPWR values across subjects for each network (relative to other networks) which resulted in better segregation of networks based on rCST and rPWR properties than CMRglc and IFCD measures (Fig. 3c, f).”

“We studied how measures of brain morphometry (i.e., cortical thickness and cortical distance), and fMRI tSNR and PET-FDG tSNR were related to IFCD, CMRglc, rPWR, and rCST. Specifically, these measures were related across subjects in 34 bilateral cortical ROIs (see Methods) in cohort-1. Across-subject mean and standard deviation in IFCD, CMRglc, rCST, and rPWR in these regions are summarized in Supplementary Table 8.”

“**Fig. 3.** CMRglc, IFCD, rPWR, and rCST of brain networks and individual differences (cohort-1, $n = 28$). (a, b) Within-subject averages (28 circles) and between-subject average (thick line) of CMRglc and IFCD for each of 10 resting-state networks²⁸ including medial visual (MV), occipital pole (OP), lateral visual (LV), default mode (DM), cerebellum (CB), sensorimotor (SM), auditory (AD), executive control (EC), right and left frontoparietal (RFP & LFP) networks. (c) Within-subject averages of CMRglc and IFCD plotted against each other (each circle represents one participant) for the networks shown in (a, b). The network colors in (c) match those shown in (a, b). (d, e) Within-subject averages (28 circles for 28 participants) and between-subject average (thick lines) of rPWR and rCST for each of the 10 resting-state networks shown in parts (a, b). (f) Within-subject averages of rPWR and rCST plotted against each other for the networks shown in (d, e). The networks are color coded in (f) based on those shown in (d, e).”

or whether they are affected by limitations of the technology itself (e.g. scanner-related issues showing stronger signal to noise ratio (SNR) in some regions compared to others. It strikes me that areas of high power may in part be conflated with areas of the brain that are more easily imaged by fMRI (e.g. visual areas and default-mode), whereas areas of low power may be harder to image due to signal dropout. Have the authors considered this, or run any analyses of whether SNR maps were significantly correlated with power/cost in these areas?

This is an important concern and we have addressed this question in a new section of Results and Discussion.

“For fMRI, tSNR (see Methods) was overall negatively associated with IFCD ($p = 0.005$) and positively with CMRglc ($p = 0.0004$) across regions (Supplementary Table 11), but not with rPWR nor with rCST. For PET-FDG, tSNR (see Methods) was negatively associated with IFCD ($p = 0.02$) and positively with CMRglc ($p < 0.0001$) (Supplementary Table 12). PET-FDG tSNR was also significantly associated with rPWR ($p = 0.02$) and rCST ($p = 0.03$), but both to a weaker extent than the associations between PET-FDG tSNR and CMRglc ($p < 0.0001$, Supplementary Table 12). While temporal pole and entorhinal regions both has the lowest tSNR in PET-FDG and fMRI, their rPWR and rCST were not associated with tSNRs (Supplementary Tables 11, 12). In fact, rPWR and rCST in none of the cortical regions were associated with tSNRs after correction for multiple comparisons. We also used PET-FDG tSNR and fMRI tSNR maps to compute SNR-based rPWR and rCST maps and performed cortical segmentation with k -means clustering ($k = 4$, Supplementary Fig. 6) and found that the clusters differed from those based on CMRglc-IFCD.”

“**Brain morphometry and temporal signal-to-noise ratio (tSNR)**... For fMRI, voxelwise tSNR was computed using the mean to standard deviation ratio of the raw fMRI time series (after motion correction and spatial normalization). For PET-FDG, voxelwise tSNR was computed using the mean to standard deviation ratio of the dynamic FDG time series (after motion correction and spatial normalization, acquired between 40–55 min). For each subject, each tSNR measure was averaged within each of the 34 bilateral ROIs.”

“However, we cannot rule out that inherent limitations in PET and MRI imaging (e.g., spatial heterogeneity in tSNR) could to some extent affect regional differences in rCST and rPWR. We observed that rPWR and rCST do not appear to amplify effects of tSNR in fMRI and PET-FDG (Supplementary Tables 11, 12 and Supplementary Fig. 6). Visual networks showed significant differences in rCST and rPWR (Fig. 3d, e; Fig. 4a), yet they had mid-to-high range tSNR in both modalities (Supplementary Tables 11, 12), suggesting that differences in rPWR and rCST in visual regions are not primarily related to measurement noise.”

I am not sure of the purpose of the the first 2 cohorts (NIH1, n = 28, and NIH, n = 7). Did the authors use these two cohorts to assess test-retest reliability? I did not see any data indicating that cohort 2 was used to verify the results of cohort 1. Were the data combined from these two cohorts? Please clarify. Similarly, how variable are these measures (cost/power) between individuals?

We are sorry for any confusion here. Cohorts are now more clearly defined in the Introduction and each section and figure cites the related cohort. The first two cohorts served different purposes. Now they are labeled cohort-vs ($n = 7$) and cohort-1 ($n = 28$). Cohort-vs was only involved with fMRI for a visual stimulation experiment at 0.5 Hz to compare sensitivity of IFCD (synchrony based) and fLAFF (amplitude based) measures of resting state activity. Cohort-1 was used for deriving rPWR and rCST (from CMRglc and IFCD) and as discussed earlier, their reliability/generalizability is now assessed in the same cohort-1 with alternative modalities (i.e., CBF and fALFF). We also addressed the question on individual differences, in an earlier comment (Fig. 3 and Supplementary Table 8).

“In cohort-vs (vs: visual stimulation; $n = 7$ healthy participants) with fMRI data only, we tested the sensitivity of IFCD to brain activity...”

“In cohort-1 ($n = 28$ healthy participants) with high-resolution PET-FDG and fMRI, voxelwise measures of relative power (rPWR; indicating high metabolic needs and observed activity relative to the rest of the brain) and relative cost (rCST; the extent to which regional metabolic needs exceeds the observed activity) were computed.”

The alcohol cohort is interesting, yet needs more explanation. Were the power/cost measures sensitive to the level of alcohol drinking, or the level of intoxication, in participants? Do the authors have any speculations about how these measures would lead to the functional outcomes experienced by long-term, heavy drinkers? Would the authors expect an association between power/cost measures and use of alternative sources of energy during intoxication?

This was a very helpful suggestion. We had previously reported effects of acute and chronic alcohol on behavioral measures in cohort-2²⁹. For acute alcohol effects, both light drinker (LD) and heavy drinker (HD) participants received a fixed amount of alcohol (0.75 g/kg) but reported significant effects in subjective experience of alcohol. For chronic effects, HD also had lower cognitive outcomes. We related these measures to CMRglc, IFCD, rCST, and rPWR and found that regional rCST and rPWR were related to these measures:

“Cohort-2 behavioral measures. In cohort-2, we previously reported behavioral measures in 23 LD participants (out of 24) and 15 HD participants (out of 16)^{24, 29}. Superficially, in both groups we found significant effects of ALC for 5 self-reported measures of feeling *sedated, dizzy, high, pleasant, and intoxication*²⁹. For summarizing subjective effects of ALC in this study, we computed the first principal component of these 5 measures (subjective-PC accounted for 68% of variance), in ALC condition across both groups. We also found HD had lower cognitive performance in 5 tasks: *Stroop (neutral, congruent, and incongruent), Symbol Digit Modalities test (SDMT), and Word Association tasks*²⁹. For summarizing individual differences in cognitive performance in HDs, we computed the first principal component of these 5 measures (cognitive-PC accounted for 50% of variance), obtained in the PLC condition in HDs.”

“Behavioral association with rPWR and rCST. In cohort-2, for subjective measures showing effects of ALC we calculated a subjective-PC (see Methods) and assessed its association with regional CMRglc, IFCD, rCST, and rPWR in both groups. We only found significant negative associations between rPWR in insula, middle and inferior temporal gyri (in the right hemisphere) and subjective experience of alcohol ($p_{FWE} < 0.01$, Supplementary Table 17). While HD performed worse in a range of cognitive tasks, individual differences in these measures in HD (cognitive-PC, see Methods) was only significantly associated with rCST in the right inferior parietal lobule ($p_{FWE} < 0.01$, Supplementary Table 18).”

“Relative to IFCD, CMRglc, and rCST, less rPWR was association with greater subjective experience of alcohol in the insula (Supplementary Table 17), suggesting that concurrent changes in (BOLD) activity and metabolism in this limbic region provide relevant markers of subjective experience of alcohol³⁰. We also found evidence that rCST (relative to IFCD, CMRglc, and rPWR) in the inferior parietal lobule was positively associated with performance in cognitive tasks in HDs (Supplementary Table 18). While activity in parietal regions have been previously related to intelligence³¹, our findings suggest that

energetic needs (i.e., rCST) in parietal areas contribute to individual differences in cognitive performance in HD.”

“More importantly, we found that rPWR and rCST (relative to IFCD and CMRglc) were significantly and distinctly related to behavioral effects of acute and chronic alcohol, further supporting their utility for capturing meaningful (and possibly unique) aspects of brain performance.”

It would be interesting to show side-by-side maps of (1) glucose metabolism (PET), (2) resting-state activity (fMRI) and (3) cost/power maps in order to directly visualize how these measures relate to one another. Showing these in the control cohorts and the alcohol cohort would help the reader visualize the potential utility of these measures for clinical purposes.

We have added Supplementary Fig. 8 which shows CMRglc, IFCD, rCST, and rPWR for the same brain cross section with same color-scaling in one LD and one HD participant from cohort-2.

“Voxelwise rCST and rPWR were estimated for the cohort-1 and cohort-2 for each participant and each condition (e.g., PLC and ALC for cohort-2, see Supplementary Fig. 8).”

References:

1. Tomasi D, Volkow ND. Functional connectivity density mapping. *Proc Natl Acad Sci U S A* **107**, 9885-9890 (2010).
2. Zang Y, Jiang T, Lu Y, He Y, Tian L. Regional homogeneity approach to fMRI data analysis. *Neuroimage* **22**, 394-400 (2004).
3. Zou Q-H, *et al.* An improved approach to detection of amplitude of low-frequency fluctuation (ALFF) for resting-state fMRI: fractional ALFF. *Journal of neuroscience methods* **172**, 137-141 (2008).
4. Biswal BB, Kannurpatti SS, Rypma B. Hemodynamic scaling of fMRI-BOLD signal: validation of low-frequency spectral amplitude as a scalability factor. *Magnetic resonance imaging* **25**, 1358-1369 (2007).
5. Tomasi D, Shokri-Kojori E, Volkow N. Temporal changes in local functional connectivity density reflect the temporal variability of the amplitude of low frequency fluctuations in gray matter. *PLoS One* **11**, e0154407 (2016).
6. Zhang XD, *et al.* Decreased coupling between functional connectivity density and amplitude of low frequency fluctuation in non-neuropsychiatric systemic lupus erythematosus: a resting-stage functional MRI study. *Molecular neurobiology* **54**, 5225-5235 (2017).

7. Vaishnavi SN, Vlassenko AG, Rundle MM, Snyder AZ, Mintun MA, Raichle ME. Regional aerobic glycolysis in the human brain. *Proceedings of the National Academy of Sciences* **107**, 17757-17762 (2010).
8. Goyal MS, Hawrylycz M, Miller JA, Snyder AZ, Raichle ME. Aerobic glycolysis in the human brain is associated with development and neotenus gene expression. *Cell metabolism* **19**, 49-57 (2014).
9. Sherwood CC, *et al.* Evolution of increased glia–neuron ratios in the human frontal cortex. *Proceedings of the National Academy of Sciences* **103**, 13606-13611 (2006).
10. DiNuzzo M, Nedergaard M. Brain energetics during the sleep–wake cycle. *Curr Opin Neurobiol* **47**, 65-72 (2017).
11. Mergenthaler P, Lindauer U, Dienel GA, Meisel A. Sugar for the brain: the role of glucose in physiological and pathological brain function. *Trends in neurosciences* **36**, 587-597 (2013).
12. Magistretti PJ. Neuron–glia metabolic coupling and plasticity. *Journal of Experimental Biology* **209**, 2304-2311 (2006).
13. Gerlach M, Ben-Shachar D, Riederer P, Youdim M. Altered brain metabolism of iron as a cause of neurodegenerative diseases? *Journal of neurochemistry* **63**, 793-807 (1994).
14. Mosconi L. Brain glucose metabolism in the early and specific diagnosis of Alzheimer’s disease. *European journal of nuclear medicine and molecular imaging* **32**, 486-510 (2005).
15. Bertolizio G, Mason L, Bissonnette B. Brain temperature: heat production, elimination and clinical relevance. *Pediatric Anesthesia* **21**, 347-358 (2011).
16. Dienel GA. Brain lactate metabolism: the discoveries and the controversies. *Journal of Cerebral Blood Flow & Metabolism* **32**, 1107-1138 (2012).
17. Volkow ND, *et al.* Acute alcohol intoxication decreases glucose metabolism but increases acetate uptake in the human brain. *Neuroimage* **64**, 277-283 (2013).
18. Wehrl HF, *et al.* Simultaneous PET-MRI reveals brain function in activated and resting state on metabolic, hemodynamic and multiple temporal scales. *Nature medicine* **19**, 1184-1189 (2013).
19. Huisman MC, *et al.* Cerebral blood flow and glucose metabolism in healthy volunteers measured using a high-resolution PET scanner. *EJNMMI research* **2**, 63 (2012).

20. Cha Y-HK, Jog MA, Kim Y-C, Chakrapani S, Kraman SM, Wang DJ. Regional correlation between resting state FDG PET and pCASL perfusion MRI. *Journal of Cerebral Blood Flow & Metabolism* **33**, 1909-1914 (2013).
21. Clarkson MJ, *et al.* A comparison of voxel and surface based cortical thickness estimation methods. *Neuroimage* **57**, 856-865 (2011).
22. Van Hoesen GW, Parvizi J, Chu C-C. Orbitofrontal cortex pathology in Alzheimer's disease. *Cerebral Cortex* **10**, 243-251 (2000).
23. Chan D, *et al.* Patterns of temporal lobe atrophy in semantic dementia and Alzheimer's disease. *Annals of neurology* **49**, 433-442 (2001).
24. Volkow ND, Wang GJ, Shokri Kojori E, Fowler JS, Benveniste H, Tomasi D. Alcohol decreases baseline brain glucose metabolism more in heavy drinkers than controls but has no effect on stimulation-induced metabolic increases. *J Neurosci* **35**, 3248-3255 (2015).
25. Volkow ND, *et al.* Low doses of alcohol substantially decrease glucose metabolism in the human brain. *NeuroImage* **29**, 295-301 (2006).
26. Valenzuela CF. Alcohol and neurotransmitter interactions. *Alcohol health and research world* **21**, 144-148 (1997).
27. Shokri-Kojori E, Tomasi D, Wiers CE, Wang GJ, Volkow ND. Alcohol affects brain functional connectivity and its coupling with behavior: greater effects in male heavy drinkers. *Mol Psychiatry*, (2016).
28. Smith SM, *et al.* Correspondence of the brain's functional architecture during activation and rest. *Proceedings of the National Academy of Sciences* **106**, 13040-13045 (2009).
29. Shokri-Kojori E, Tomasi D, Wiers CE, Wang GJ, Volkow ND. Alcohol affects brain functional connectivity and its coupling with behavior: greater effects in male heavy drinkers. *Molecular Psychiatry* **22**, 1185 (2016).
30. Bjork JM, Gilman JM. The effects of acute alcohol administration on the human brain: insights from neuroimaging. *Neuropharmacology* **84**, 101-110 (2014).
31. Lee KH, *et al.* Neural correlates of superior intelligence: stronger recruitment of posterior parietal cortex. *Neuroimage* **29**, 578-586 (2006).

Reviewers' comments:

Reviewer #1 (Remarks to the Author):

The authors have addressed my concerns from the previous version, I recommend a minor revision considering the following points:

1. Is the stimulation data (which lacks a metabolic measure such as glucose or blood flow) necessary to have in this paper? It seems like it could be its own paper instead. The submitted paper could be shortened by instead citing existing papers comparing IFCD and fALFF to functional activation, e.g. Thompson, G. J., et al. (2016). *Brain Connect* 6(6): 435-447.
2. Discussion of the utility the new measures of rPWR and rCST versus existing metrics should be more up-front in the introduction and discussion. E.g. discussion of Figure 3, also how new measures do not correlate with anatomy as strongly as glucose metabolism, etc.
3. Some acronym are not defined in the paper, e.g. COMET, PC. This should be added. A table of acronyms would also be helpful.
4. Different color scales are used on different plots in Figure 1, unless there is a good reason they should be the same colors, to help with visual comparison.
5. "decreased the kurtosis (uniformity)" should probably be "decreased the kurtosis (increased uniformity)"

Reviewer #2 (Remarks to the Author):

This is a revision of manuscript where I substantially questioned the justification and validity of the imaging parameters of the first submission. This was also the main concern of the two other reviewers. I am therefore puzzled that the authors did not attempt to better substantiate their interpretation of cost/power but simply renamed them to "relative CST/PWR".

The two parameters cost/power do not homogeneously or substantially vary across the majority of cortical regions. Fig 2g/h indicates a highly similar cortical distribution and this is justified by the difference map in Fig 4a. Fig 4a indicates that only two major midline structures do significantly differ in rCST/PWR which are medial visual/parietal and temporal cortices. All frontal, sensori, motor, and parietal regions cannot be associated with having either predominantly rCST or rPWR respectively. This can also be seen in Fig 3d/e which shows that 5 out of the 10 networks are highly variable in rCST/rPWR, but the first three are all visual networks. The majority of cortex would be, however, represented in the later 5 networks that basically show no variance in any of the 2 measures. Importantly, Fig 3b shows that IFCD only varies in the 3 visual networks but basically all other major networks are located at the lower bottom of IFCD.

In my first revision, I was therefore questioning that the raw distribution of CMRGlu and IFCD is suitable to justify an equal distribution across the cortex and the 4 virtual quadrants. Rotating and normalizing the data does not compensate for that problem. My concern that the majority of the voxels are simply located at the lower left quadrant has not been resolved as Fig 2b and Fig 4c do not seem to match (I understand that Fig 2c is just a model).

Interestingly, the authors deleted the "confusing figure Suppl Fig1" from the initial submission that was intended to show the necessary relationship between CMRGlu and IFCD which is now replaced with a new Suppl Fig 1a that shows no significant correlation and is used as an argument for generalizability of rPWR/CST.

Other parameters during analysis are fully arbitrary. Why are raw data rotated by 45°? Why do r-thresholds for IFCD-computation vary across cohorts?

Overall, I have the impression that analyses and interpretation are highly selective and speculative but the current imaging measures do not justify the general interpretation of (relative) power or

cost related to neural activity across the entire human cortex.

Reviewer #3 (Remarks to the Author):

The authors did a commendable job of responding to all reviewer comments, and have included several new tables/figures that help with interpretation of the data. Though this manuscript has limitations, the authors have addressed most of these in the review process.

Reviewers' comments:

Reviewer #1 (Remarks to the Author):

The authors have addressed my concerns from the previous version, I recommend a minor revision considering the following points:

We thank the reviewer for the helpful feedback and the opportunity to further improve the manuscript.

1. Is the stimulation data (which lacks a metabolic measure such as glucose or blood flow) necessary to have in this paper? It seems like it could be its own paper instead. The submitted paper could be shortened by instead citing existing papers comparing IFCD and fALFF to functional activation, e.g. Thompson, G. J., et al. (2016). Brain Connect 6(6): 435-447.

As the reviewer suggested, we removed the visual stimulation data but added relevant citations.

“In this relation, indirect measures of neuronal activity such as fMRI measures of functional connectivity^{1,2} or MRS measures of glutamatergic function³, have been associated with regional brain glucose metabolism, wherein high and low neuronal activity demand were associated with high and low metabolic supply, respectively.”

“Synchrony-based and amplitude-based measures appear to capture overlapping and nonoverlapping aspects of brain activity. For example, they have shown good correspondence at rest⁴ and between different subject groups⁵. Our results indicated an excellent agreement between IFCD-based and fALFF-based rPWR estimates and between IFCD-based and fALFF-based rCST estimates, across major brain networks (Supplementary Fig. 6c and Supplementary Fig. 7c). However, there are indications that synchrony-based measures may be more sensitive to changes in functional activity (e.g., eyes open versus closed) than amplitude-based measures⁶. These observations are consistent with the relatively high spatial spread of neuronal activity with respect to the stimulation locus^{7,8} that could be captured with synchrony-based measures of brain activity.”

2. Discussion of the utility the new measures of rPWR and rCST versus existing metrics should be more up-front in the introduction and discussion. E.g. discussion of Figure 3, also how new measures do not correlate with anatomy as strongly as glucose metabolism, etc.

We further clarified the utility of the rPWR and rCST in the introduction and emphasized on Fig. 2 findings (former Fig. 3) and the anatomical correlations in the first paragraph of discussion.

“There are marked regional differences in glucose metabolism^{9,10} and in fMRI measures of brain activity^{11,12,13} during resting state that are positively associated across regions^{1,2}. However, without accounting for underlying brain activity, regional differences in glucose metabolism are hard to interpret. Interestingly, the level of correspondence between glucose metabolism and neuroglial activity has been considered an important marker of functional specialization of brain regions and networks¹⁴, and could be helpful for inferring use of alternative energy sources or different metabolic pathways. To further study this, here we proposed an approach to quantify match and mismatch between measured

metabolic supply and the observed level of activity across the brain and assessed whether this quantification is relevant for studying distinct energetic characteristics of brain regions and networks.”

We also introduced a metric to quantify the level of network segregation:

“We quantified the level of segregation of brain networks in the two spaces (i.e., IFCD-CMRglc versus rPWR-rCST) using a network segregation index (NSI). Specifically, in a given two-dimensional space and for each brain network, NSI was defined as the ratio of the average of between network distances to the average of within network distances (see Methods). As expected, in the rPWR-rCST space we found significantly higher NSIs (NSI: $M = 4.34$, $SD = 1.46$, Fig. 2f) than in the IFCD-CMRglc space (NSI: $M = 1.07$, $SD = 0.66$, Fig. 2c) across the 10 networks ($p = 0.002$, Wilcoxon signed rank test).”

“When considering individual differences in IFCD and CMRglc (Fig. 2c), we found that brain networks were more segregated based on rPWR and rCST (Fig. 2f) than based on IFCD and CMRglc (Fig. 2c), further supporting the relevance of our approach for studying brain functional specialization. Across cortical regions, rPWR and rCST were not consistently associated with cortical thickness (Supplementary Table 9) and cortical distance (Supplementary Table 10) as were CMRglu and IFCD, suggesting that these anatomical features are differently related to energetic characteristics of different brain regions. Additionally, we found that rPWR and rCST indices were distinctly sensitive to the effects of acute and chronic alcohol exposure on the brain and behavior.”

3. Some acronym are not defined in the paper, e.g. COMET, PC. This should be added. A table of acronyms would also be helpful.

Accordingly, we clarified the acronyms throughout the text and added a table of acronyms (Table 1).

“Please refer to Table 1 for the list of acronyms.”

“...we calculated a subjective-PC (PC: principal component)...”

“Fig. 4a–d shows the connectivity-metabolism (COMET) maps highlighting alcohol-related changes in IFCD and CMRglc...”

“and CBF (with perfusion weighted imaging: PWI) (see Methods)...”

4. Different color scales are used on different plots in Figure 1, unless there is a good reason they should be the same colors, to help with visual comparison.

The justification for using different color scales in the (new) Fig. 1 (particularly, in parts g and h) is added to the figure caption. In addition, we added new supplementary figures (Supplementary Figs. 2 and 3) which show IFCD, CMRglc, rPWR, and rCST, in the slice-view format but with the same color scales.

“...Note the color scale in (g) resembles that used for the hypothetical voxels along the rPWR axis in (d) and the color scale in (h) resembles that used for the hypothetical voxels along the rCST axis in (f).”

5. "decreased the kurtosis (uniformity)" should probably be "decreased the kurtosis (increased uniformity)"

Thank you for bringing this to our attention. The sentence is now corrected.

"...and decreased the kurtosis (increased uniformity)..."

Reviewer #2 (Remarks to the Author):

This is a revision of manuscript where I substantially questioned the justification and validity of the imaging parameters of the first submission. This was also the main concern of the two other reviewers. I am therefore puzzled that the authors did not attempt to better substantiate their interpretation of cost/power but simply renamed them to "relative CST/PWR".

We thank the reviewer for the careful attention and the critical comments. Our intention has been to thoroughly address all the reviewer's concerns. We renamed the metrics based on reviewer 1's previous comment to avoid confusion with other measures in the literature. In addition, we now better substantiate the interpretation of rPWR and rCST in the Introduction and Discussion.

"There are marked regional differences in glucose metabolism^{9, 10} and in fMRI measures of brain activity^{11, 12, 13} during resting state that are positively associated across regions^{1, 2}. However, without accounting for underlying brain activity, regional differences in glucose metabolism are hard to interpret. Interestingly, the level of correspondence between glucose metabolism and neuroglial activity has been considered an important marker of functional specialization of brain regions and networks¹⁴, and could be helpful for inferring use of alternative energy sources or different metabolic pathways. To further study this, here we proposed an approach to quantify match and mismatch between measured metabolic supply and the observed level of activity across the brain and assessed whether this quantification is relevant for studying distinct energetic characteristics of brain regions and networks. For this purpose, we measured cerebral metabolic rate of glucose (CMRglc, indexed by ¹⁸F-fluorodeoxyglucose; PET-FDG, see Methods) and synchronous fluctuations in the blood oxygenation level (BOLD; measured by fMRI and indexed by local functional connectivity density: lFCD, see Methods) during resting state. We studied two main (unit-free and generalizable) dimensions of associations. The first dimension captured the positive association between glucose utilization and neuroglial activity. This dimension was labeled relative power (rPWR) and represented the level of concurrent metabolic need and observed activity, relative to the rest of the brain. The second dimension captured the deviation between glucose utilization and neuroglial activity. This dimension was labeled relative cost (rCST) and represented the extent to which glucose metabolic needs exceed (or fall behind) the observed activity, relative to the rest of the brain. As in principal component analysis (PCA), when there is complete correspondence between measured neuroglial activity and glucose utilization across regions, all the common variance will be accounted for by the rPWR dimension. Yet, more deviation between observed neuroglial activity and glucose utilization^{14, 15, 20, 21} (i.e., disproportional neuroglial activity and glucose utilization) across regions, would result in higher variance accounted for by the rCST dimension (Fig. 1, see Methods)."

“To quantify the relative changes in the association between two modalities, here we proposed two novel metrics of rPWR and rCST that are unit-free and are generalizable to measures of brain activity and energy supply. These metrics quantify how well two modalities that measure aspects of activity demand (such as IFCD) and aspects metabolic supply (such as CMRglc) are concurrently high or low (i.e., rPWR) in a specific region relative to the rest of the brain, and to what extent one exceeds the other (i.e., rCST) in a specific region relative to the rest of the brain. From another perspective, rPWR could be thought of as an index of concurrent *intensity* of the two modalities, while rCST is an index of *mismatch* between the two modalities.”

The two parameters cost/power do not homogeneously or substantially vary across the majority of cortical regions. Fig 2g/h indicates a highly similar cortical distribution and this is justified by the difference map in Fig 4a. Fig 4a indicates that only two major midline structures do significantly differ in rCST/PWR which are medial visual/parietal and temporal cortices. All frontal, sensori, motor, and parietal regions cannot be associated with having either predominantly rCST or rPWR respectively. This can also be seen in Fig 3d/e which shows that 5 out of the 10 networks are highly variable in rCST/rPWR, but the first three are all visual networks. The majority of cortex would be, however, represented in the later 5 networks that basically show no variance in any of the 2 measures.

We have made several important clarifications to address this. We have clarified in the Methods that rPWR and rCST are orthogonal variables (they are not anti-correlated), thus rPWR and rCST estimates are not required to be always different throughout the cortex. Accordingly, it was not a requirement for regions to be “associated with having either predominantly rCST or rPWR, respectively.” Also, for better visual comparison, we provided a slice-view of rPWR and rCST as Supplementary Figs. 2&3 (Fig. 2 also includes the former Supplementary Fig. 7). At our cluster-size corrected threshold, majority of the cortical areas had significantly different rPWR than rCST (> 54% of gray matter voxels), including many regions within frontal, sensorimotor, and parietal cortices. It was not critical to identify all regions that have different rPWR than rCST, thus in Fig. 3a (former Fig. 4a), we only highlighted extreme differences in the rCST-rPWR contrast by using a stringent threshold ($p < 10^{-7}$, $p_{FWE} < 0.01$, corrected at the voxel level). This was also the case for Supplementary Tables 1 and 2. In addition, all networks presented in Fig. 2 (former Fig. 3) are significantly different in their rPWR and rCST ($p < 0.02$).

“In addition, all networks had significantly different rPWR than rCST ($p < 0.02$).”

“This 45° counterclockwise rotation is equivalent to performing a PCA on two positively correlated variables that are, mean and variance normalized (resulting in an equal contribution of imaging parameters into rPWR and rCST). It is important to note that rPWR and rCST variables are orthogonal, thus it is possible for regions to have high (or low) rPWR and rCST at the same time, while other regions may be high in rCST and low in rPWR or vice versa. While rPWR captures most of the common variance, rCST captures the remainder of the common variance between $z(\log(\text{IFCD}))$ and $z(\text{CMRglc})$. This is evident in Supplementary Fig. 3 histograms, showing higher variability across regions in rPWR (capturing most of the variance) relative to rCST.”

“The contrast analysis indicated that most of the cortex had different rCST than rPWR (approximately 54% of gray matter voxels, $p_{FWE} < 0.01$, cluster-size corrected). However, the most pronounced differences were in the visual cortices with higher rPWR than rCST and in the limbic and temporal regions with higher rCST than rPWR (see Fig. 2a and Supplementary Table 3 for highlighted differences). There were also notable differences in rPWR and rCST between regions (Fig. 2, Supplementary Table 8).”

“For highlighting the most extreme differences, we used a conservative threshold of $p_{FWE} < 0.01$ at the voxel level ($|t| > 6$). At the normal threshold ($p_{FWE} < 0.01$, cluster-size corrected), majority of the cortex (> 54%) showed differences in rCST versus rPWR (see Supplementary Table 3 for more details).”

“When indicated, we used a more stringent threshold (corrected at the voxel level, $p_{FWE} < 0.01$, $|t| > 6$) to further guide summarizing large effects.”

Importantly, Fig 3b shows that IFCD only varies in the 3 visual networks but basically all other major networks are located at the lower bottom of IFCD.

While the 10 studied networks represented important gray matter areas¹⁶, they do not represent the entire cortex. In the new Supplementary Figs. 1a and 2 and Supplementary Table 8, we highlighted the variability of IFCD across the entire cortex. For the reviewer reference, we also performed a one sample *t*-test on z-scored log(IFCD) which not only highlighted a range of visual regions but also included large segments of parietal, cingulate, insular, frontal, and thalamic regions for having significantly higher IFCD than the rest of the brain (approximately 20% of the gray matter voxels, $p_{FWE} < 0.01$, Rebuttal Fig. 1). Interestingly (and contrastingly), the 3 visual networks with marked IFCD differences, had comparable CMRglc (Fig. 2a, b), which further supports the relevance of studying these regional differences in observed activity and level of metabolic supply, under a unified framework of rPWR and rCST.

Rebuttal Figure 1. Regions with significantly higher log(IFCD) than the rest of the brain ($p_{FWE} < 0.01$, cluster-size corrected). The MNI z-coordinate is displayed next to each slice.

In my first revision, I was therefore questioning that the raw distribution of CMRGlc and IFCD is suitable to justify an equal distribution across the cortex and the 4 virtual quadrants. Rotating and normalizing the data does not compensate for that problem. My concern that the majority of the voxels are simply located at the lower left quadrant has not been resolved as Fig 2b and Fig 4c do not seem to match (I understand that Fig 2c is just a model).

We have added new figures to further delineate regional variability in IFCD and CMRGlc.

“Spatial distributions of IFCD and CMRGlc are highlighted in Fig. 1a–c and Supplementary Figs. 1a & 2, delineating regional variability...”

“**Supplementary Figure 1.** Correlation between CMRGlc and IFCD across 34 bilateral cortical parcellations (see Methods) ($r(32) = 0.42, p = 0.01$) (a), and between rCST and rPWR ($r(32) = 0.03, p = 0.87$) (b) in cohort-1. Across the 34 bilateral regions the coefficient of variation of IFCD was 23% and coefficient of variation of CMRGlc was 18%.”

Importantly, our analysis does not assume an equal distribution of CMRGlc and IFCD across the 4 quadrants. We have clarified this in the text:

“While Fig. 1e show a hypothetical model with relatively equal distribution of voxels along the 4 identified quadrants and with no apparent association between the measures, a positive association between IFCD (measure of activity) and CMRGlc (measure of metabolic supply) would result in more voxels being associated with high and low-rPWR quadrants than high and low-rCST quadrants.”

Furthermore, this has been clarified in the Fig. 3c caption. In addition, there is no mismatch in the data used to generate Fig. 1b and Fig. 3c (former Fig. 2b and Fig. 3c). Relative to the average $\log(\text{IFCD})$ and CMRGlc shown in Fig. 3c, 30.5% and 36% of voxels fall in the high and low-rPWR quadrants, respectively.

“It is important to note that having a higher percentage of voxels associated with high and low-rPWR quadrants, is consistent with $\log(\text{IFCD})$ and CMRGlc being positively correlated (see Fig. 1b).”

Interestingly, the authors deleted the “confusing figure Suppl Fig1” from the initial submission that was intended to show the necessary relationship between CMRGlc and IFCD which is now replaced with a new Suppl Fig 1a that shows no significant correlation and is used as an argument for generalizability of rPWR/CST.

We deleted the previous Supplementary Fig. 1 based on reviewer’s previous comment: “The authors report a strong linear correlation between $\log(\text{IFCD})$ and CMRGlc (Suppl Fig 1) which I found not interpretable with respect to the raw data presented in Fig 1B of the main manuscript. The heat plot in Fig 1B rather suggests a very confined and heavy aggregation of voxels towards the lower left end of the graph with almost no variance along the $\log(\text{IFCD})$ axis. With respect to the slope in Suppl Fig 1 this linear correlation is hardly justified when plotting onto the heatmap data.” However, Fig. 1b and Fig. 3c both show the same data (as the previous Supplementary Fig. 1) but without the linear fit. We agree that showing the positive association between $\log(\text{IFCD})$ and CMRGlc is important. This is also visible in Fig. 3c where most voxels fall along the high and low-rPWR quadrants than the high and low-rCST quadrants (as defined in Fig. 1e).

The Supplementary Fig. 1a in the revised paper showed a positive, yet nonsignificant, association between $\log(\text{IFCD})$ and CMRglc across the 10 networks. In Supplementary Fig. 1a we now show the significant association between IFCD and CMRglc across 34 bilateral cortical regions that cover the entire cortex. We also report the positive Pearson correlation coefficient for data in Fig. 1b.

We did not use lack of significant correlation as an evidence for generalizability. We discussed that lack of strong correlation across the 10 networks suggests heterogeneity of these associations. For generalizability or measures, for example, we performed additional analyses using measures of amplitude of low frequency fluctuations and cerebral blood flow to assess how well rPWR and rCST calculated based on these measures, correspond to when they are calculated using IFCD and CMRglc .

“Spatial distributions of IFCD and CMRglc are highlighted in Fig. 1a–c and Supplementary Figs. 1a & 2, delineating regional variability in these measures while showing an overall positive association between IFCD and CMRglc across the brain regions (cohort-1, $n = 28$).”

“A two-dimensional histogram of $\log(\text{IFCD})$ versus cerebral metabolic rate of glucose (CMRglc) highlighting the frequency of $\log(\text{IFCD})$ and CMRglc association pairs ($r = 0.44$, $p < 0.00001$, 139269 voxels).”

Other parameters during analysis are fully arbitrary. Why are raw data rotated by 45°? Why do r-thresholds for IFCD-computation vary across cohorts?

As discussed earlier, we have clarified in the Methods that the 45° counterclockwise rotation is equivalent to performing a PCA on mean and variance normalized $\log(\text{IFCD})$ and CMRglc , ensuring equal contribution of both measures to the primary (rPWR) and secondary (rCST) dimensions of associations between $\log(\text{IFCD})$ and CMRglc .

The manuscript now reports one primary IFCD threshold of $r = 0.6$ which is consistent with prior literature using similar fMRI temporal resolution^{6, 11, 17}. As the reviewer indicated before, we tested the effect of IFCD threshold on rPWR and rCST measures and found excellent agreement between rPWR (and rCST) estimated with $r = 0.6$ and $r = 0.4$ IFCD thresholds (Supplementary Figs. 6d, 7d). In the visual stimulation data, we had used a lower threshold of $r = 0.5$ for fMRI data which was acquired with a high temporal resolution of $\text{TR} = 0.385$ s (relative to $\text{TR} = 1.5$ s). As the reviewer 1 suggested, this experiment is now removed from the manuscript. In addition, we have estimated rPWR and rCST using fALFF instead of IFCD (which does not require a threshold) (see Supplementary Figs. 6, 7).

In addition, we added a new measure to quantify the level of network segregation in the IFCD-CMRglc and rPWR-rCST spaces.

“We quantified the level of segregation of brain networks in the two spaces (i.e., IFCD-CMRglc versus rPWR-rCST) using a network segregation index (NSI). Specifically, in a given two-dimensional space and for each brain network, NSI was defined as the ratio of the average of between network distances to the average of within network distances (see Methods). As expected, in the rPWR-rCST space we found significantly higher NSIs (NSI: $M = 4.34$, $SD = 1.46$, Fig. 2f) than in the IFCD-CMRglc space (NSI: $M = 1.07$, $SD = 0.66$, Fig. 2c) across the 10 networks ($p = 0.002$, Wilcoxon signed rank test).”

“Network segregation index (NSI). This measure was defined to index how well a brain network (or a region) is segregated from other brain networks (or regions), when individual participant values for all the networks are plotted in a two-dimensional space (e.g., Fig. 2c versus Fig. 2f). For this purpose, three parameters were calculated. For a given network (N_i), center of mass (CM_i) coordinates were calculating by averaging of each dimension across participants. For N_i , average within network distance ($AWND_i$) was calculated by averaging the Euclidian distance of participants data points from the CM_i . For N_i , average between network distance ($ABND_i$) was calculated by averaging the Euclidian distance from CM_i to the CMs of the rest of the networks. Finally, NSI could be defined as following, which is expected to have a F -distribution:

$$NSI_i = \frac{ABND_i}{AWND_i}.$$

Overall, I have the impression that analyses and interpretation are highly selective and speculative but the current imaging measures do not justify the general interpretation of (relative) power or cost related to neural activity across the entire human cortex.

We have carefully considered all specific concerns raised by the reviewers and have accordingly clarified or justified our approach. We have further delineated regional variability in the imaging measures and their relative distribution, have clarified underlying assumptions of rPWR and rCST parameters, and added an index to quantify the level of network segregation in different spaces. We have highlighted that IFCD and CMRglc only capture a specific aspect of brain activity demand and metabolic supply but provided evidence that studying the regional variations in the associations between these imaging measures is important for functional characterization of brain regions and networks. We are happy to further consider any specific concerns.

Reviewer #3 (Remarks to the Author):

The authors did a commendable job of responding to all reviewer comments, and have included several new tables/figures that help with interpretation of the data. Though this manuscript has limitations, the authors have addressed most of these in the review process.

Once again, we thank the reviewer for the constructive feedback and we hope that future work would further address limitations.

References:

1. Tomasi D, Wang GJ, Volkow ND. Energetic cost of brain functional connectivity. *Proceedings of the National Academy of Sciences of the United States of America* **110**, 13642-13647 (2013).
2. Aiello M, *et al.* Relationship between simultaneously acquired resting-state regional cerebral glucose metabolism and functional MRI: a PET/MR hybrid scanner study. *NeuroImage* **113**, 111-121 (2015).
3. Sibson NR, Dhankhar A, Mason GF, Rothman DL, Behar KL, Shulman RG. Stoichiometric coupling of brain glucose metabolism and glutamatergic neuronal activity. *Proceedings of the National Academy of Sciences* **95**, 316-321 (1998).
4. Tomasi D, Shokri-Kojori E, Volkow N. Temporal changes in local functional connectivity density reflect the temporal variability of the amplitude of low frequency fluctuations in gray matter. *PLoS One* **11**, e0154407 (2016).
5. Zhang XD, *et al.* Decreased coupling between functional connectivity density and amplitude of low frequency fluctuation in non-neuropsychiatric systemic lupus erythematosus: a resting-stage functional MRI study. *Molecular neurobiology* **54**, 5225-5235 (2017).
6. Thompson GJ, Riedl V, Grimmer T, Drzezga A, Herman P, Hyder F. The whole-brain “global” signal from resting state fMRI as a potential biomarker of quantitative state changes in glucose metabolism. *Brain connectivity* **6**, 435-447 (2016).
7. Grinvald A, Lieke EE, Frostig RD, Hildesheim R. Cortical point-spread function and long-range lateral interactions revealed by real-time optical imaging of macaque monkey primary visual cortex. *Journal of Neuroscience* **14**, 2545-2568 (1994).
8. Rosenbaum R, Smith MA, Kohn A, Rubin JE, Doiron B. The spatial structure of correlated neuronal variability. *Nature Neuroscience*, (2016).
9. Gur R, *et al.* Sex differences in regional cerebral glucose metabolism during a resting state. *Science* **267**, 528-531 (1995).
10. Kleinridders A, *et al.* Regional differences in brain glucose metabolism determined by imaging mass spectrometry. *Molecular metabolism* **12**, 113-121 (2018).
11. Tomasi D, Volkow ND. Functional connectivity density mapping. *Proc Natl Acad Sci U S A* **107**, 9885-9890 (2010).
12. Zuo X-N, *et al.* The oscillating brain: complex and reliable. *Neuroimage* **49**, 1432-1445 (2010).

13. Wang G-Z, *et al.* Correspondence between resting-state activity and brain gene expression. *Neuron* **88**, 659-666 (2015).
14. Vaishnavi SN, Vlassenko AG, Rundle MM, Snyder AZ, Mintun MA, Raichle ME. Regional aerobic glycolysis in the human brain. *Proceedings of the National Academy of Sciences* **107**, 17757-17762 (2010).
15. Wang J, *et al.* Oxidation of ethanol in the rat brain and effects associated with chronic ethanol exposure. *Proc Natl Acad Sci U S A* **110**, 14444-14449 (2013).
16. Smith SM, *et al.* Correspondence of the brain's functional architecture during activation and rest. *Proceedings of the National Academy of Sciences* **106**, 13040-13045 (2009).
17. Liu C, *et al.* Aberrant patterns of local and long-range functional connectivity densities in schizophrenia. *Oncotarget* **8**, 48196 (2017).

REVIEWERS' COMMENTS:

Reviewer #1 (Remarks to the Author):

The authors have addressed my minor comments from the last revision, and done a good work in addressing Reviewer 2's comments. In particular I like the network segregation method. I recommend acceptance of this manuscript.

Reviewer #2

[Was not available to re-review.]

Reviewer #3

[In comments to the editor, indicates that the authors had adequately responded to Reviewer #2 concerns, and that she/he is happy to recommend publication at this stage.]